# ARTreeFormer: A faster attention-based auto-regressive model for phylogenetic inference

## Abstract

Probabilistic modeling of the combinatorially explosive tree topology space has posed a significant challenge in phylogenetic inference. Previous approaches often necessitate pre-sampled tree topologies, limiting their modeling capability to a subset of the entire tree space. A recent advancement is ARTree, a deep autoregressive model that offers unrestricted distributions for tree topologies. However, the repetitive computations of topological node embeddings via Dirichlet energy minimization and the message passing over all the nodes can be expensive, which may hinder its application to data sets with many species. This paper proposes ARTreeFormer, a novel approach that harnesses attention mechanisms to accelerate ARTree. By introducing attention-based recurrent node embeddings, ARTreeFormer allows the reuse of node embeddings from preceding ordinal tree topologies and fast vectorized computation as well. This, together with a local message passing scheme, significantly improves the computation speed of ARTree while maintaining great approximation performance. We demonstrate the effectiveness and efficiency of our method on a benchmark of challenging real data phylogenetic inference problems.

## 1 Introduction

Unraveling the evolutionary relationships among species stands as a core problem in the field of computational biology. This complex task, called *phylogenetic inference*, is abstracted as the statistical inference on the hypothesis of shared history, i.e., *phylogenetic trees*, based on collected molecular sequences (e.g., DNA, RNA) of the species of interest and a model of evolution. Phylogenetic inference finds its diverse applications ranging from genomic epidemiology (Dudas et al., 2017; du Plessis et al., 2021; Attwood et al., 2022) to the study of conservation genetics (DeSalle & Amato, 2004). Classical approaches for phylogenetic inference includes maximum likelihood (Felsenstein, 1981), maximum parsimony (Fitch, 1971), and Bayesian approaches (Yang & Rannala, 1997; Mau et al., 1999; Larget & Simon, 1999), etc. Nevertheless, phylogenetic inference remains a hard challenge partially due to the combinatorially explosive size ($(2N - 5)!!$ for unrooted bifurcating trees with $N$ species) of the phylogenetic tree topology space (Whidden & Matsen IV, 2015; Dinh et al., 2017), which makes many common principles in phylogenetics, e.g., maximum likelihood and maximum parsimony, to be NP-hard problems (Chor & Tuller, 2005; Day, 1987).

Recently, the prosperous development of machine learning provides an effective and innovative approach to phylogenetic inference, and many efforts have been made for expressive probabilistic modeling of the tree topologies (Höhna & Drummond, 2012; Larget, 2013; Zhang & Matsen IV, 2018; Xie & Zhang, 2023). A notable example among them is ARTree (Xie & Zhang, 2023), which provides a rich family of tree topology distributions and achieves state-of-the-art performance on benchmark data sets. Given a specific order on the leaf nodes (also called the taxa order), ARTree generates a tree topology by sequentially adding a new leaf node to an edge of the current subtree topology at a time, according to an edge decision distribution modeled by graph neural networks (GNNs), until all the leaf nodes have been added. Compared with previous methods such as conditional clade distribution (CCD) (Larget, 2013) and subsplit Bayesian networks (SBNs) (Zhang & Matsen IV, 2018), an important advantage of ARTree is that it enjoys unconfined support over the entire tree topology space. However, to compute the edge decision distribution in each leaf node addition step, ARTree requires expensive repetitive computations of topological node embeddings based on

Dirichlet energy minimization and message passing over all the nodes, making it prohibitive for phylogenetic inference for large numbers of species, as observed in Xie & Zhang (2023).

With the emergence of transformer architectures (Vaswani et al., 2017) in recent years, numerous studies have demonstrated their promising performances in graph representation learning (Yun et al., 2019; Ying et al., 2021; Rampášek et al., 2022). In this paper, we propose ARTreeFormer, which enables faster ancestral sampling and probability evaluation compared to ARTree, leveraging transformer architectures. More specifically, we substitute the time-consuming node embedding module with a learnable recurrent node embedding module, which computes the node embeddings for the newly added nodes using the attention-based graph-level information of the preceding subtree topologies. To further reduce the computational cost of the message passing module, we design an attention-based local message passing scheme that only updates the embedding vectors of the neighbors of those newly added nodes. Moreover, unlike ARTree, all these modules in ARTreeFormer can be easily vectorized across different tree topologies and different nodes. This way, ARTreeFormer is capable of generating/evaluating a batch of tree topologies simultaneously, while ARTree can only do this one by one. In experiments, we demonstrate that ARTreeFormer achieves comparable results but around $10\times$ generation speed and $3\times$ training speed than ARTree on a benchmark of challenging maximum parsimony, tree topology density estimation, and variational Bayesian phylogenetic inference problems.

## 2 BACKGROUND

**Phylogenetic posterior**   The common structure for describing evolutionary history is a phylogenetic tree, which consists of a bifurcating tree topology $\tau$ and the associated non-negative branch lengths $\boldsymbol{q}$. The tree topology $\tau$, which contains leaf nodes for the observed species and internal nodes for the unobserved ancestor species, represents the evolutionary relationship among these species. A tree topology can be either rooted or unrooted. In this paper, we only discuss unrooted tree topologies, but the proposed method can be easily adapted to rooted tree topologies. The branch lengths $\boldsymbol{q}$ quantify the evolutionary intensity along the edges on $\tau$. An edge is called a pendant edge if it connects one leaf node to an internal node.

Each leaf node on $\tau$ corresponds to a species with an observed biological sequence (e.g., DNA, RNA, protein). Let $\boldsymbol{Y} = \{Y_1, \ldots, Y_M\} \in \Omega^{N \times M}$ be the observed sequences (with characters in $\Omega$) of $M$ sites over $N$ species. A continuous-time Markov chain is commonly assumed to model the transition probabilities of the characters along the edges of a phylogenetic tree (Felsenstein, 2004). Under the assumption that different sites evolve independently and identically, the likelihood of observing sequences $\boldsymbol{Y}$ given a phylogenetic tree $(\tau, \boldsymbol{q})$ takes the form

$$p(\boldsymbol{Y}|\tau, \boldsymbol{q}) = \prod_{i=1}^{M} \sum_{a^i} \eta(a_r^i) \prod_{(u,v) \in E} P_{a_u^i a_v^i}(q_{uv}), \tag{1}$$

where $a^i$ ranges over all extensions of $Y_i$ to the internal nodes with $a_u^i$ being the character assignment of node $u$ ($r$ represents the root node), $E$ is the set of edges of $\tau$, $q_{uv}$ is the branch length of the edge $(u, v) \in E$, $P_{jk}(q)$ is the transition probability from character $j$ to $k$ through an edge of length $q$, and $\eta$ is the stationary distribution of the Markov chain. Assuming a prior distribution $p(\tau, \boldsymbol{q})$ on phylogenetic trees, Bayesian phylogenetic inference then amounts to properly estimating the posterior distribution

$$p(\tau, \boldsymbol{q}|\boldsymbol{Y}) = \frac{p(\boldsymbol{Y}|\tau, \boldsymbol{q})p(\tau, \boldsymbol{q})}{p(\boldsymbol{Y})} \propto p(\boldsymbol{Y}|\tau, \boldsymbol{q})p(\tau, \boldsymbol{q}). \tag{2}$$

**Variational Bayesian phylogenetic inference**   By positing a phylogenetic variational family $Q_{\boldsymbol{\phi}, \boldsymbol{\psi}}(\tau, \boldsymbol{q}) = Q_{\boldsymbol{\phi}}(\tau)Q_{\boldsymbol{\psi}}(\boldsymbol{q}|\tau)$ as the product of a tree topology model $Q_{\boldsymbol{\phi}}(\tau)$ and a conditional branch length model $Q_{\boldsymbol{\psi}}(\boldsymbol{q}|\tau)$, variational Bayesian phylogenetic inference (VBPI) converts the inference problem (2) into an optimization problem. More specifically, VBPI seeks the best variational approximation by maximizing the following multi-sample lower bound

$$L^K(\boldsymbol{\phi}, \boldsymbol{\psi}) = \mathbb{E}_{Q_{\boldsymbol{\phi}, \boldsymbol{\psi}}(\tau^{1:K}, \boldsymbol{q}^{1:K})} \log \left( \frac{1}{K} \sum_{i=1}^{K} \frac{p(\boldsymbol{Y}|\tau^i, \boldsymbol{q}^i)p(\tau^i, \boldsymbol{q}^i)}{Q_{\boldsymbol{\phi}}(\tau^i)Q_{\boldsymbol{\psi}}(\boldsymbol{q}^i|\tau^i)} \right), \tag{3}$$

where $Q_{\phi,\psi}(\tau^{1:K}, \boldsymbol{q}^{1:K}) = \prod_{i=1}^{K} Q_{\phi,\psi}(\tau^i, \boldsymbol{q}^i)$. In addition to the likelihood $p(\boldsymbol{Y}, \tau, \boldsymbol{q})$ in the numerator of equation (3), one may also consider the parsimony score defined as the minimum number of character-state changes among all possible sequence assignments for internal nodes, i.e.,

$$\mathcal{P}(\tau; \boldsymbol{Y}) = \sum_{i=1}^{M} \min_{a^i} \sum_{(u,v) \in E} \mathbb{I}(a_u^i \neq a_v^i), \tag{4}$$

where the notations are the same as in equation (1) (Zhou et al., 2024). The parsimony score $\mathcal{P}(\tau; \boldsymbol{Y})$ can be efficiently evaluated by the Fitch algorithm (Fitch, 1971) in linear time.

The tree topology model $Q_\phi(\tau)$ can take subsplit Bayesian networks (SBNs) (Zhang & Matsen IV, 2018) which rely on subsplit support estimation for parametrization, or ARTree (Xie & Zhang, 2023) which is an autoregressive model using graph neural networks (GNNs) that provides distributions over the entire tree topology space. A diagonal lognormal distribution is commonly used for the branch length model $Q_\psi(\boldsymbol{q}|\tau)$ whose locations and scales are parameterized with heuristic features (Zhang & Matsen IV, 2019; Zhang, 2020) or learnable topological features (Zhang, 2023). More details about VBPI can be found in Appendix C.

**ARTree for tree topology generation**   As an autoregressive model for tree topology generation, ARTree (Xie & Zhang, 2023) decomposes a tree topology into a sequence of leaf node addition decisions and models the involved conditional probabilities with GNNs. The corresponding tree topology generating process can be described as follows. Let $\mathcal{X} = \{x_1, \ldots, x_N\}$ be the set of leaf nodes with a pre-defined order. The generating procedure starts with a simple tree topology $\tau_3 = (V_3, E_3)$ that has the first three nodes $\{x_1, x_2, x_3\}$ as the leaf nodes (which is unique), and keeps adding new leaf nodes according to the following rule. Given an intermediate tree topology $\tau_n = (V_n, E_n)$ that has the first $n < N$ elements in $\mathcal{X}$ as the leaf nodes, i.e., an *ordinal tree topology* of rank $n$ as defined in Xie & Zhang (2023), a probability vector $q_n \in \mathbb{R}^{|E_n|}$ over the edge set $E_n$ is first computed via GNNs. Then, an edge $e_n \in E_n$ is sampled according to $q_n$ and the next leaf node $x_{n+1}$ is attached to it to form an ordinal tree topology $\tau_{n+1}$. This procedure will continue until all the $N$ leaf nodes are added. Although a pre-defined leaf node order is required, Xie & Zhang (2023) shows that the performance of ARTree exhibits negligible dependency on this leaf node order. See more details on ARTree in Appendix B.

## 3    PROPOSED METHOD

Although ARTree enjoys unconfined support over the entire tree topology space and provides a more flexible family of variational distributions, it suffers from expensive computation costs (see Appendix E in Xie & Zhang (2023)) which makes it prohibitive for phylogenetic inference when the number of species is large. In this section, we first discuss the computational cost of ARTree and then describe how it can be accelerated using attention-based techniques.

### 3.1    COMPUTATIONAL COST OF ARTREE

In the $n$-th step of leaf node addition, ARTree includes the node embedding module and message passing module for computing the edge decision distribution, detailed below. Throughout this section, we use "node embeddings" (with dimension $N$) for the node information before message passing and "node features" (with dimension $d$) for those in and after message passing.

**Node embedding module**   The topological node embeddings $\{f_n(u) \in \mathbb{R}^N | u \in V_n\}$ of an ordinal tree topology $\tau_n = (V_n, E_n)$ in Xie & Zhang (2023) are obtained by first assigning one-hot encodings to the leaf nodes and then minimizing the *global Dirichlet energy*

$$\ell(f_n, \tau_n) := \sum_{(u,v) \in E_n} \|f_n(u) - f_n(v)\|^2, \tag{5}$$

which is typically done by the two-pass algorithm (Zhang, 2023) (Algorithm 2 in Appendix B). This algorithm requires a traversal over a tree topology, which cannot be efficiently vectorized across different nodes due to its serial nature. Moreover, this cannot be vectorized across different trees

since this traversal depends on a specific tree topology shape. The complexity of computing the topological node embeddings is $O(Nn)$. Finally, a multi-layer perceptron (MLP) is applied to all the node embeddings to obtain the node features with dimension $d$ enrolled in the computation of the following modules.

**Message passing module**    Assume the the initial node features are $\{f_n^0(u) \in \mathbb{R}^d | u \in V_n\}$ at the beginning of message passing. In the $l$-th round, these node features are updated by aggregating the information from their neighborhoods through

$$m_n^l(u, v) = F_{\text{message}}^l(f_n^l(u), f_n^l(v)), \tag{6a}$$

$$f_n^{l+1}(v) = F_{\text{updating}}^l \left( \{m_n^l(u, v); u \in \mathcal{N}(v)\} \right), \tag{6b}$$

where the $l$-th message function $F_{\text{message}}^l$ and updating function $F_{\text{updating}}^l$ consist of MLPs. These two functions are applied to the features of all the nodes on $\tau_n$, called global message passing by us, which require $O(nd^2)$ operations and is computationally inefficient especially when the number of leaf nodes is large.

Figure 2 (left) demonstrates the run time and floating points operations (FLOPs) of ARTree as the number of leaf nodes $N$ varies. As $N$ increases, the total run time of ARTree grows rapidly and the node embedding module dominates the total time ($\approx 65\%$), which makes ARTree prohibitive when the number of leaf nodes is large. The reason behind this is that compared to other modules, the node embedding module can not be easily vectorized w.r.t. different tree topologies and different nodes, resulting in great computational inefficiency (more than 10 seconds for generating 60 100-leaf trees).

## 3.2 ATTENTION-BASED EDGE DECISION DISTRIBUTION

In this section, we propose ARTreeFormer, which introduces attention-based recurrent node embeddings and a local message passing scheme to accelerate the training and sampling in ARTree. Denote the node features for the ordinal tree topology $\tau_n = (V_n, E_n)$ at the $n$-th step of the generating process as $\{f_n(u) \in \mathbb{R}^d | u \in V_n\} =: \mathcal{F}_n \in \mathbb{R}^{(2n-3) \times d}$. We start from the smallest ordinal tree topology $\tau_3$ by setting $f_3(x_1), f_3(x_2), f_3(x_3) \in \mathbb{R}^d$ to be learnable parameters. In what follows, we present our approach for modeling the edge decision distribution at the $n$-th step.

**Recurrent node embedding module**    Instead of re-computing the topological node embeddings which wastes the information from the previously generated tree topologies, ARTreeFormer tries to learn the node embeddings from this information with a deep graph model. To achieve this, it first uses the attention mechanism to compute a graph representation vector $r_n \in \mathbb{R}^d$, i.e.,

$$\bar{r}_n = F_{\text{graph}}(q_n, \mathcal{F}_n, \mathcal{F}_n), \tag{7a}$$

$$r_n = R_{\text{graph}}(\bar{r}_n), \tag{7b}$$

where $F_{\text{graph}}$ is the graph pooling function implemented as a multi-head attention block (Vaswani et al., 2017), $R_{\text{graph}}$ is the graph readout function implemented as a 2-layer MLP, and $q_n \in \mathbb{R}^d$ is a learnable query vector. Here, the multi-head attention block $M = \text{MHA}(Q, K, V)$ is defined as

$$H_i = \text{softmax} \left( \frac{(QW_i^Q)(KW_i^K)'}{\sqrt{d/h}} \right) \cdot (VW_i^V), \tag{8a}$$

$$M = \text{CONCAT}(H_1, \ldots, H_h) W^O, \tag{8b}$$

where $W_i^Q, W_i^K, W_i^V \in \mathbb{R}^{d \times \frac{d}{h}}$ and $W^O \in \mathbb{R}^{d \times d}$ are learnable matrices, $h$ is the number of heads, and CONCAT is the concatenation operator along the node feature axis. Intuitively, we have used a global vector $q_n$ to query all the node features and obtained a representation vector $r_n$ for the whole tree topology $\tau_n$. We emphasize that equation (7) enjoys time complexity $O(nd + d^2)$ instead of the $O(n^2 d + nd^2)$ of common multi-head attention blocks, as $q_n$ is a one-dimensional vector.

We now compute the edge decision distribution to decide where to add the next leaf node, similarly to ARTree. To incorporate global information into the edge decision, we utilize the global representation vector $r_n$ to compute the edge features. Concretely, the feature of an edge $e = (u, v)$ is formed by

$$p_n(e) = F_{\text{edge}} \left( \{f_n(u), f_n(v)\} \right), \tag{9a}$$

$$r_n(e) = R_{\text{edge}} \left( \text{CONCAT}(p_n(e), r_n) + b_n \right), \tag{9b}$$

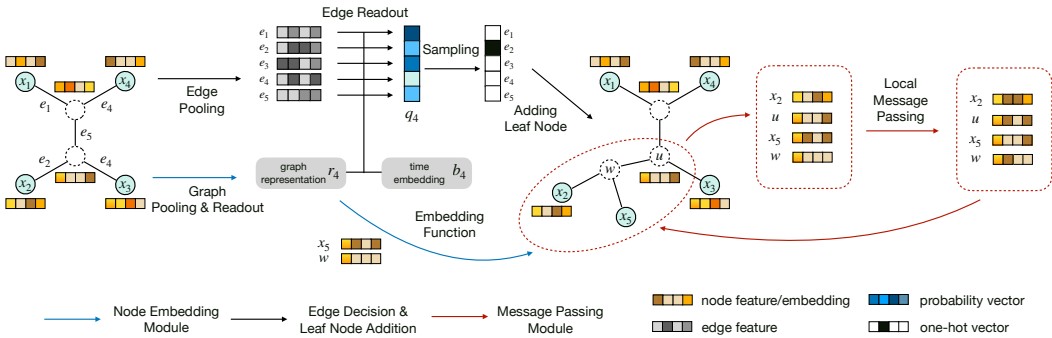

Figure 1: An illustration of ARTreeFormer for growing an ordinal tree topology $\tau_4$ of rank 4 to an ordinal tree topology $\tau_5$ of rank 5.

where $F_{\text{edge}}$ is an invariant edge pooling function implemented as an elementwise maximum operator, $R_{\text{edge}}$ is the edge readout function implemented as a 2-layer MLP with scalar output, and $b_n$ is the sinusoidal positional embedding (Vaswani et al., 2017) of the time step $n$. Then one can calculate the edge decision distribution $Q_\phi(\cdot|e_{<n})$ using

$$Q_\phi(\cdot|e_{<n}) = \text{Discrete}(\alpha_n), \ \alpha_n = \text{softmax}\left([r_n(e)]_{e \in E_n}\right), \tag{10}$$

and grow $\tau_n$ to $\tau_{n+1}$ by attaching the next leaf node $x_{n+1}$ to the sampled edge (Algorithm 3).

We then make use of the graph representation vector $r_n$ to compute the embedding vectors of the newly added nodes, while keeping the embedding vectors of other nodes unchanged. In ARTreeFormer, the node embedding for newly added leaf node $x_{n+1}$ is given by

$$f_n(x_{n+1}) = F_{\text{emb}}(r_n), \tag{11}$$

where the embedding function $F_{\text{emb}}$ is set to be a 2-layer MLP. Note that we still use a subscript $n$ for the node embeddings $f_n$ as one additional message passing module is needed to form $f_{n+1}$. To assign an embedding vector to the newly added internal node $w$ which is connected to $x_{n+1}$ through a pendant edge, we minimize the *local Dirichlet energy* of $w$ defined as

$$\ell(f_n, \tau_{n+1}, w) := \sum_{(u,w) \in E_{n+1}} \|f_n(u) - f_n(w)\|^2 \tag{12}$$

in contrast to minimizing the global Dirichlet energy (5) in ARTree. This way, the embedding vector for the node $w$ is just the arithmetic mean of the embedding vectors of its neighbors.

**Local message passing module** To further reduce the computation cost caused by applying the message passing module in equation (6) to all the nodes, ARTreeFormer adopts a local updating scheme in the neighborhood of the newly added internal node $w$, similarly to Han et al. (2023). Specifically, letting $\mathcal{F}_n^{\text{local}} := \{f_n(u)|u \in \mathcal{N}(w)\} \in \mathbb{R}^{4 \times d}$, the local message passing scheme takes the form

$$\bar{\mathcal{F}}_n^{\text{local}} = F_{\text{message}}\left(\mathcal{F}_n^{\text{local}}, \mathcal{F}_n^{\text{local}}, \mathcal{F}_n^{\text{local}}\right), \tag{13}$$

where $\bar{\mathcal{F}}_n^{\text{local}} = \{\bar{f}_n(u)|u \in \mathcal{N}(w)\}$ is the updated local node features and the message function $F_{\text{message}}$ is a multi-head attention block described in equation (8) whose time complexity is $O(d^2)$. Here, the computational complexity of the message passing module is downscaled by a factor of $n$ compared to ARTree since only local node features are updated. Finally, the node features $f_{n+1}$ for the tree topology $\tau_{n+1}$ are given by

$$f_{n+1}(u) = \begin{cases} \bar{f}_n(u), & u \in \mathcal{N}(w), \\ f_n(u), & u \notin \mathcal{N}(w). \end{cases} \tag{14}$$

The above two modules circularly continue until an ordinal tree topology of $N$, $\tau_N$, is constructed, whose ARTreeFormer-based probability is defined as

$$Q_\phi(\tau_N) = \prod_{n=3}^{N-1} Q_\phi(e_n|e_{<n}), \tag{15}$$

where $\phi$ are the learnable parameters and $Q_\phi(e_n|e_{<n})$ is defined in equation (10).

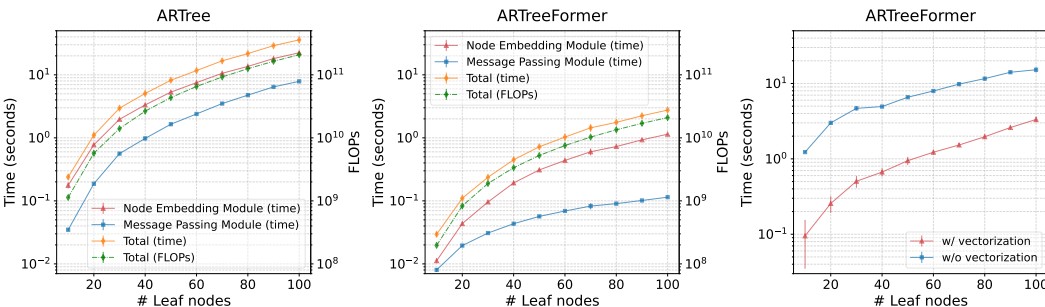

Figure 2: **Left**: Runtime and FLOPs for generating 100 tree topologies using ARTree. **Middle**: Runtime and FLOPs for generating 100 tree topologies using ARTreeFormer. **Right**: The runtime of ARTreeFormer for generating 100 tree topologies with or without vectorization. All tests are run on a single 2.4 GHz CPU.

---

**Algorithm 1:** Growing an ordinal tree topology $\tau_n$ to $\tau_{n+1}$ with ARTreeFormer

---

**Input:** An ordinal tree topology $\tau_n = (V_n, E_n)$ with $n$ leaf nodes; the node features $\mathcal{F}_n$ of $\tau_n$.
**Output:** An ordinal tree topology $\tau_{n+1} = (V_{n+1}, E_{n+1})$ with $n + 1$ leaf nodes; the node
        features $\mathcal{F}_{n+1}$ of $\tau_{n+1}$.
\# *node embedding module*
Compute the graph representation vector $r_n$ using $\mathcal{F}_n$ as in equation (7);
Compute the edge features $r_n(e)$ for all $e \in E_n$ with $\mathcal{F}_n$ and $r_n$ as in equation (9);
Compute the edge decision distribution $Q_\phi(\cdot|e_{<n})$ with the edge features as in equation (10);
Sample an edge decision $e_n$ from $Q_\phi(\cdot|e_{<n})$ and grow $\tau_n$ to $\tau_{n+1}$ as described in Algorithm 3;
Compute the features of the newly added nodes by minimizing local Dirichlet energy (12);
\# *message passing module*
Update the local node features using the attention mechanism as described in equation (13);
Obtain $\mathcal{F}_{n+1}$ by replacing the local node features in $\mathcal{F}_n$ with the update ones, as in equation (14).

---

Compared to ARTree, the greatly improved computational efficiency of ARTreeFormer mainly comes from two aspects. **First**, the learnable node embedding module as well as local Dirichlet energy minimization in ARTreeFormer can be easily vectorized across different tree topologies and different nodes, since they do not rely on the specific tree topology shape nor require traversals over the tree topologies. **Second**, the local message passing in ARTreeFormer avoids applying deep models to all the node features, in contrast with the global message passing in ARTree.

| Model | Node embedding | |
|---|---|---|
| | Compl. | Vec. Compl. |
| ARTree | $O(N^3)$ | $O(N^{2+\alpha})$ |
| ARTreeFormer | $O(N^2 d + N d^2)$ | $O(N^{1+\alpha} d^\alpha + N d^{2\alpha})$ |
| Model | Message passing | |
| | Compl. | Vec. Compl. |
| ARTree | $O(N^2 d^2)$ | $O(N^{1+\alpha} d^{2\alpha})$ |
| ARTreeFormer | $O(N d^2)$ | $O(N d^{2\alpha})$ |

Table 1: Computational complexity (Compl.) and computational complexity with vectorized operations (Vec. Compl.) of generating an $N$-leaf tree topology. $\alpha \in (0, 1)$ refers to the accelerated order of vectorized linear operations.

Figure 2 (left, middle) shows that the run time and FLOPs of ARTreeFormer are significantly reduced to 10% of ARTree. To further verify the vectorization capability of ARTreeFormer, we compare the runtime for generating tree topologies with or without vectorization (i.e., simultaneously or sequentially) in Figure 2 (right), where vectorization greatly improves computational efficiency. Summing up all the involved complexities for $n = 3, \ldots, N$ gives Table 1. Although $\alpha$ can be small in practice (i.e., fast computation of batched tensors), the complexity of ARTree's node embedding module is still higher than $O(N^2)$, while those of other modules are reduced to approximately equal to or less than $O(N)$. This validates the observation that the topological node embedding dominates the computation time. Further discussion on Table 1 can be found in Appendix B.3.

Several previous efforts (Yun et al., 2019; Ying et al., 2021; Rampášek et al., 2022) have demonstrated the power of transformers for graph representation learning. Especially, Han et al. (2023) considers variational inference on graphs with a transformer-based autoregressive generative model. Our approach differs from them in the following aspects. First, the learnable node embedding based on the attention mechanism is novel and overcomes the non-vectorizable bottleneck of ARTree. Second, we incorporate message passing and local Dirichlet energy minimization within the neighborhood structure, specifically designed for phylogenetic trees. Third, adapting graph techniques to phylogenetic trees is not straightforward and requires careful design, and we are the first to show that this simplified attention-based architecture exhibits strong approximation capacity with considerably reduced computational cost. More discussions on the related works in the field of phylogenetic inference are deferred to Appendix A.

## 4 EXPERIMENTS

In this section, we demonstrate the effectiveness and efficiency of ARTreeFormer on three benchmark tasks: maximum parsimony, tree topology density estimation (TDE), and variational Bayesian phylogenetic inference (VBPI). Although the pre-selected leaf node order in ARTreeFormer may not be related to the relationships among species, this evolutionary information is already contained in the training data set (for TDE) or the target posterior distribution (for maximum parsimony and VBPI), and thus can be learned by ARTreeFormer. Noting that the main contribution of ARTreeFormer is improving the tree topology model, we select the first two tasks because they only learn the tree topology distribution and can better demonstrate the superiority of ARTreeFormer. The third task, VBPI, is selected as a standard benchmark task for Bayesian phylogenetic inference and evaluates how well ARTreeFormer collaborates with a branch length model. It should be emphasized that we mainly pay attention to the computational efficiency improvement of ARTreeFormer and only expect it to attain similar accuracy with baseline methods.

**Experimental setup** For TDE and VBPI, we perform experiments on eight data sets which we will call DS1-8. These data sets, consisting of sequences from 27 to 64 eukaryote species with 378 to 2520 site observations, are commonly used to benchmark phylogenetic MCMC methods (Hedges et al., 1990; Garey et al., 1996; Yang & Yoder, 2003; Henk et al., 2003; Lakner et al., 2008; Zhang & Blackwell, 2001; Yoder & Yang, 2004; Rossman et al., 2001; Höhna & Drummond, 2012; Larget, 2013; Whidden & Matsen IV, 2015). For the Bayesian setting in MrBayes runs (Ronquist et al., 2012) (an MCMC software for Bayesian phylogenetic inference), we assume a uniform prior on the tree topologies, an i.i.d. exponential prior $\mathrm{Exp}(10)$ on branch lengths, and the simple JC substitution model (Jukes et al., 1969). We use the same ARTreeFormer structure across all the data sets for all three experiments. Specifically, we set the dimension of node features to $d = 100$, following Xie & Zhang (2023). The number of heads in all the multi-head attention blocks is set to $h = 4$. All the activation functions for MLPs are exponential linear units (ELUs) (Clevert et al., 2015). We add a layer normalization block after each linear layer in MLPs and before each multi-head attention block (Xiong et al., 2020). We also add a residual block after the multi-head attention block in the message passing step, which is standard in transformers. The taxa order is set to the lexicographical order of the corresponding species names. All models are implemented in PyTorch (Paszke et al., 2019) and optimized with the Adam (Kingma & Ba, 2015) optimizer. All the experiments are run on an Intel Xeon Platinum 8358 processor. The learning rate for ARTreeFormer is set to 0.0001 in all the experiments.

### 4.1 MAXIMUM PARSIMONY PROBLEM

We first test the performance of ARTreeFormer on solving the maximum parsimonious problem. We reformulate this problem as a Bayesian inference task with the target distribution $P(\tau) = \exp(-\mathcal{P}(\tau, \boldsymbol{Y}))/Z$, where $\mathcal{P}(\tau, \boldsymbol{Y})$ is the parsimony score defined in equation (4) and $Z = \sum_{\tau} \exp(-\mathcal{P}(\tau, \boldsymbol{Y}))$ is the normalizing constant. To fit a variational distribution $Q_{\phi}(\tau)$, we maximize the following (annealed) multi-sample lower bound ($K = 10$) in the $t$-th iteration

$$\mathcal{L}(\phi) = \mathbb{E}_{Q_{\phi}(\tau^{1:K})} \log \left( \frac{1}{K} \sum_{i=1}^{K} \frac{\exp\left(-\beta_t \mathcal{P}(\tau_i, \boldsymbol{Y})\right)}{Q_{\phi}(\tau_i)} \right), \tag{16}$$

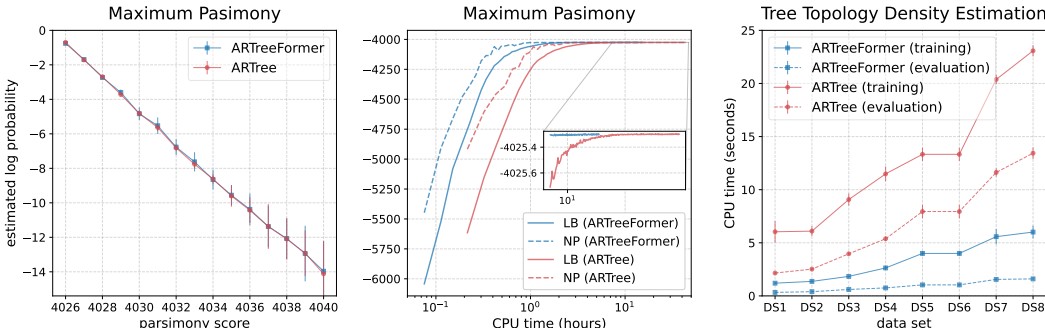

Figure 3: Performances of ARTree and ARTreeFormer on various tasks. **Left**: The estimated log probability $\log Q(\tau)$ versus the parsimony score $\mathcal{P}(\tau, \boldsymbol{Y})$ on DS1. For different tree topologies with the same parsimony score, the mean of the estimated log probabilities is plotted as a dot with the standard deviation as the error bar. **Middle**: The 10-sample lower bound (LB) and the negative parsimony score (NP) as a function of the CPU time on DS1. **Right**: The training time (per 10 iterations) and evaluation time (per computing the probabilities of 100 tree topologies) of ARTree and ARTreeFormer across eight benchmark data sets for TDE. The results are averaged over 100 runs with the standard deviation as the error bar.

Table 2: KL divergences to the ground truth of different methods across eight benchmark data sets. The "Sampled trees" column shows the numbers of unique tree topologies in the training sets. The "GT trees" column shows the numbers of unique tree topologies in the ground truth. The results are averaged over 10 replicates. The results of SBN-EM, SBN-EM-$\alpha$ are from Zhang & Matsen IV (2018), and the results of SBN-SGA and ARTree are from Xie & Zhang (2023). For each data set, the best result is marked in **black bold font** and the second best result is marked in **brown bold font**.

| Data set | # Taxa | # Sites | Sampled trees | GT trees | KL divergence to ground truth | | | | |
| --- | --- | --- | --- | --- | --- | --- | --- | --- | --- |
| | | | | | SBN-EM | SBN-EM-$\alpha$ | SBN-SGA | ARTree | ARTreeFormer |
| DS1 | 27 | 1949 | 1228 | 2784 | 0.0136 | 0.0130 | 0.0504 | **0.0045** | **0.0067** |
| DS2 | 29 | 2520 | 7 | 42 | 0.0199 | 0.0128 | 0.0118 | **0.0097** | **0.0102** |
| DS3 | 36 | 1812 | 43 | 351 | 0.1243 | 0.0882 | 0.0922 | **0.0548** | **0.0777** |
| DS4 | 41 | 1137 | 828 | 11505 | 0.0763 | 0.0637 | 0.0739 | **0.0299** | **0.0320** |
| DS5 | 50 | 378 | 33752 | 1516877 | 0.8599 | 0.8218 | 0.8044 | **0.6266** | **0.6681** |
| DS6 | 50 | 1133 | 35407 | 809765 | 0.3016 | 0.2786 | 0.2674 | **0.2360** | **0.2478** |
| DS7 | 59 | 1824 | 1125 | 11525 | 0.0483 | 0.0399 | 0.0301 | **0.0191** | **0.0271** |
| DS8 | 64 | 1008 | 3067 | 82162 | 0.1415 | 0.1236 | 0.1177 | **0.0741** | **0.0667** |

where $Q_\phi(\tau^{1:K}) = \prod_{i=1}^{K} Q_\phi(\tau^i)$ and $\beta_t$ is the annealing schedule. We set $\beta_t = \min\{t/200000, 1\}$ and collect the results after 400000 parameter updates. We use the VIMCO estimator (Mnih & Rezende, 2016) to estimate the stochastic gradients of $\mathcal{L}(\phi)$.

The first two plots in Figure 3 show the performances of different methods for the maximum parsimony problem on DS1. We run the state-of-the-art parsimony analysis software PAUP* (Swofford, 2003) to form the ground truth, which contains tree topologies with parsimony scores ranging from 4040 to the optimal score 4026. The left plot of Figure 3 shows that both ARTreeFormer and ARTree can identify the most parsimonious tree topology found by PAUP* and provide comparably accurate posterior estimates. In the middle plot of Figure 3, the horizontal gap between two curves reflects the ratio of times needed to reach the same lower bound or negative parsimony score. We see that ARTreeFormer is around three times faster than ARTree.

### 4.2 TREE TOPOLOGY DENSITY ESTIMATION

We further investigate the capacity of ARTreeFormer for modeling tree topologies on the TDE task. To construct the training data set, we run MrBayes (Ronquist et al., 2012) on each data set with 10 replicates of 4 chains and 8 runs until the runs have ASDSF (the standard convergence criteria used in MrBayes) less than 0.01 or a maximum of 100 million iterations, collect the samples every 100 iterations, and discard the first 25%, following Zhang & Matsen IV (2018). The ground truth

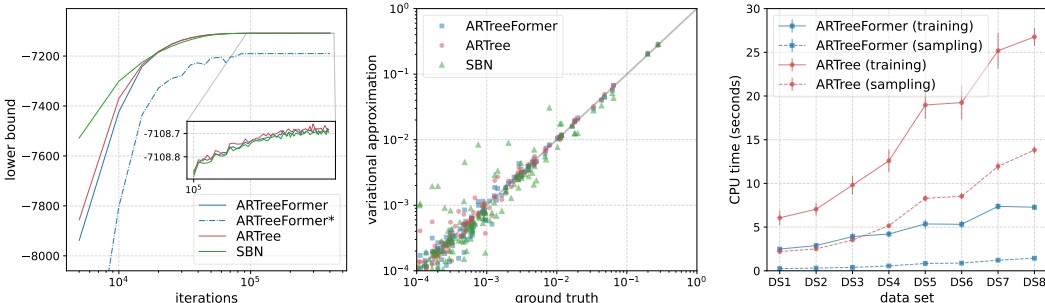

Figure 4: Performances of different methods for VBPI. **Left**: the 10-sample lower bound as a function of the number of iterations on DS1. The ARTreeFormer* refers to the de-attention version of ARTreeFormer which does not contain multi-head attention in forming recurrent node embeddings and message passing. **Middle**: the variational approximation v.s. the ground truth of the marginal distribution of tree topologies on DS1. **Right**: Training time (per 10 iterations) and sampling time (per sampling 100 tree topologies) across different data sets. The results are averaged over 100 runs with the standard deviation as the error bar.

distributions are obtained from 10 extremely long single-chain MrBayes runs, each for one billion iterations, where the samples are collected every 1000 iterations, with the first 25% discarded as burn-in. We train ARTreeFormer via maximum likelihood estimation using stochastic gradient ascent. We compare ARTreeFormer to ARTree and SBN baselines: i) for SBN-EM and SBN-EM-$\alpha$, the SBN model is optimized using the expectation-maximization (EM) algorithm, as done in Zhang & Matsen IV (2018); ii) for SBN-SGA and ARTree, the corresponding models are fitted via stochastic gradient ascent, similar to ARTreeFormer. For SBN-SGA, ARTree, and ARTreeFormer, the results are collected after 200000 parameter updates with a batch size of 10.

The right plot in Figure 3 shows a significant reduction in the training time and evaluation time of ARTreeFormer compared to ARTree on DS1-8. The KL divergences between the ground truth and the probability estimation are reported in Table 2. Although ARTreeFormer has a simplified model structure for node features, it performs on par or better than ARTree, and consistently outperforms the SBN baselines, across all data sets. See the probability estimation on individual tree topologies and an ablation study about the hyperparameters in Appendix D.

### 4.3 VARIATIONAL BAYESIAN PHYLOGENETIC INFERENCE

Our last experiment is on VBPI, where we examine the performance of ARTreeFormer on tree topology posterior approximation (Section 2). Following Xie & Zhang (2023), we use the following annealed unnormalized posterior as our target at the $t$-th iteration

$$p(\tau, \boldsymbol{q}|\boldsymbol{Y}, \beta_i) \propto p(\boldsymbol{Y}|\tau, \boldsymbol{q})^{\beta_t} p(\tau, \boldsymbol{q}), \tag{17}$$

where $\beta_t = \min\{1, 0.001 + t/200000\}$ is the annealing schedule. We set $K = 10$ for the multi-sample lower bound (3) and use the VIMCO estimator (Mnih & Rezende, 2016) and reparametrization trick (Kingma & Welling, 2014) to obtain the gradient estimates for the tree topology parameters and the branch lengths parameters respectively. The results are collected after 400000 parameter updates. To be fair, for all three VBPI-based methods (VBPI-SBN, VBPI-ARTree, and VBPI-ARTreeFormer), we use the same branch length model that is parametrized by GNNs with edge convolutional operator and learnable topological features as done in Zhang (2023). We also consider three alternative approaches ($\phi$-CSMC (Koptagel et al., 2022), GeoPhy (Mimori & Hamada, 2023)) that provide unconfined tree topology distributions and one MCMC based method (MrBayes) as baselines.

The left plot in Figure 4 shows the lower bound as a function of the number of iterations on DS1. We see that although ARTreeFormer converges slower than SBN and ARTree at the beginning, it quickly catches up and reaches a similar lower bound in the end. The result of ARTreeFormer* demonstrates the effectiveness of the attention mechanism in modeling the tree topologies. The middle plot in Figure 4 shows that both ARTree and ARTreeFormer can provide accurate variational approximations to the ground truth posterior of tree topologies, and both of them outperform SBNs by a large margin.

Table 3: Marginal likelihood estimates (in units of nats) of different methods across eight benchmark data sets for Bayesian phylogenetic inference. The marginal likelihood estimates for ARTreeFormer are obtained by importance sampling with 1000 particles from the variational approximation and are averaged over 100 independent runs with standard deviation in the brackets. The results of MrBayes SS which serve as the ground truth are from Zhang & Matsen IV (2019). The results of other methods are reported in their original papers.

| Data set | DS1 | DS2 | DS3 | DS4 | DS5 | DS6 | DS7 | DS8 |
|---|---|---|---|---|---|---|---|---|
| # Taxa | 27 | 29 | 36 | 41 | 50 | 50 | 59 | 64 |
| # Sites | 1949 | 2520 | 1812 | 1137 | 378 | 1133 | 1824 | 1008 |
| $\phi$-CSMC (Koptagel et al., 2022) | -7290.36(7.23) | -30568.49(31.34) | -33798.06(6.62) | -13582.24(35.08) | -8367.51(8.87) | -7013.83(16.99) | N/A | -9209.18(18.03) |
| GeoPhy (Mimori & Hamada, 2023) | -7111.55(0.07) | -26368.44(0.13) | -33735.85(0.12) | -13337.42(1.32) | -8233.89(6.63) | -6733.91(0.57) | -37350.77(11.74) | -8660.48(0.78) |
| VBPI-SBN (Zhang, 2023) | -7108.41(0.14) | -26367.73(0.07) | -33735.12(0.09) | **-13329.94(0.19)** | -8214.64(0.38) | **-6724.37(0.40)** | -37332.04(0.26) | -8650.65(0.45) |
| VBPI-ARTree (Xie & Zhang, 2023) | -7108.41(0.19) | **-26367.71(0.07)** | **-33735.09(0.09)** | **-13329.94(0.17)** | **-8214.59(0.34)** | **-6724.37(0.46)** | **-37331.95(0.27)** | **-8650.61(0.48)** |
| **VBPI-ARTreeFormer (ours)** | **-7108.40(0.21)** | **-26367.71(0.09)** | **-33735.09(0.08)** | **-13329.94(0.20)** | -8214.63(0.40) | -6725.09(0.44) | -37331.96(0.26) | -8650.62(0.49) |
| MrBayes SS (Xie et al., 2011) | -7108.42(0.18) | -26367.57(0.48) | -33735.44(0.50) | -13330.06(0.54) | -8214.51(0.28) | -6724.07(0.86) | -37332.76(2.42) | -8649.88(1.75) |

In the right plot of Figure 4, we see that the computation time of ARTreeFormer is substantially reduced compared to ARTree. This reduction is especially evident for sampling time since it does not include the branch length generation, likelihood computation, and backpropagation.

Table 3 shows the marginal likelihood estimates obtained by different methods on DS1-8, including the results of the stepping-stone (SS) method (Xie et al., 2011), which is one of the state-of-the-art sampling based methods for marginal likelihood estimation. We find that VBPI-ARTreeFormer provides comparable estimates to VBPI-SBN and VBPI-ARTree. Compared to other VBPI variants, the methodological and computational superiority of ARTreeFormer is mainly reflected by its unconfined support (compared to SBN) and faster computation speed (compared to ARTree). All VBPI variants perform on par with SS, while the other baselines ($\phi$-CSMC, GeoPhy) tend to provide underestimated results. We also note that the standard deviations of ARTreeFormer can be larger than ARTree and SBN which can be partially attributed to the potentially less accurate approximation. Regarding the efficiency-accuracy trade-off, for relatively small data sets, the simplified architecture in ARTreeformer is enough to maintain or even surpass the performance of ARTree; for larger data sets, a performance drop in approximation accuracy may be observed. We also provide an ablation study on the hyperparameters and more information on the memory and time consumption of different methods for VBPI in Appendix E. Finally, it is worth noting that VBPI-mixture (Molén et al., 2024; Hotti et al., 2024) can provide a better marginal likelihood approximation by employing mixtures of tree models as the variational family.

## 5 CONCLUSION

In this work, we presented ARTreeFormer, a variant of ARTree that leverages the attention mechanism to accelerate the autoregressive modeling of tree topologies in phylogenetic inference. In contrast to ARTree, which involves repetitive computations for Dirichlet energy minimization based node embeddings during the tree topology generating process, ARTreeFormer reused the graph features of preceding tree topologies by introducing an attention-based learnable recurrent node embedding module. This, together with a local message passing scheme, greatly reduced the computational cost and enabled vectorized computation over different nodes and tree topologies as well. Experiments on various phylogenetic inference problems showed that ARTreeFormer is significantly faster than ARTree in training and evaluation while performing comparably in terms of approximation accuracy.

Phylogenetic inference provides critical insights for making informed public health decisions, particularly during pandemics. Developing efficient Bayesian phylogenetic inference algorithms that can deliver accurate posterior estimates in a timely manner is therefore of immense value, with the potential to save countless lives. The commonly used MCMC methods tend to be slow and often requires long runs to generate high quality samples. In contrast, VI approaches hold significant promise due to their optimization-based framework. For example, VI methods have been used for rapid analysis of pandemic-scale data (e.g., SARS-CoV-2 genomes) to provide accurate estimates of epidemiologically relevant quantities that can be corroborated via alternative public health data sources (Ki & Terhorst, 2022). We expect more efficient VI approaches for Bayesian phylogenetics and associated software to be developed in the near future, further advancing this critical field.

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

## A    RELATED WORKS

The most common approach for Bayesian phylogenetic inference is Markov chain Monte Carlo (MCMC), which relies on random walks to explore the tree space, e.g., MrBayes (Ronquist et al., 2012). Although MCMC methods are often considered state-of-the-art in this field, they often exhibit low exploration efficiency and require extremely long runs to deliver accurate posterior estimates (Whidden & Matsen IV, 2015; Zhang & Matsen IV, 2024).

Another approach is variational inference (VI) which requires a variational family over the phylogenetic trees. Besides VBPI introduced in Section 2, there exist other VI methods. VaiPhy (Koptagel et al., 2022) approximates the posterior of multifurcating trees with a novel sequential tree topology sampler based on maximum spanning trees. GeoPhy (Mimori & Hamada, 2023) models the tree topology distribution through a mapping from continuous distributions over the leaf nodes to tree topologies via the Neighbor-Joining (NJ) algorithm (Saitou & Nei, 1987).

As a classical tool in Bayesian statistics, sequential Monte Carlo (SMC) (Bouchard-Côté et al., 2012) and its variant combinatorial SMC (CSMC) (Wang et al., 2015) propose to sample tree topologies

through subtree merging and resampling steps for Bayesian phylogenetic inference. Moretti et al. (2021) employs a learnable proposal distribution based on CSMC and optimizes it within a variational framework. Koptagel et al. (2022) further makes use of the parameters of VaiPhy to design the proposal distribution for sampling bifurcating trees ($\phi$-CSMC). The subtree merging operation in SMC based methods is also the core idea of PhyloGFN (Zhou et al., 2024), which instead treats the merging choices as actions within the GFlowNet (Bengio et al., 2021) framework and optimizes the trajectory balance objective (Malkin et al., 2022).

## B   DETAILS OF ARTREE

### B.1   TREE TOPOLOGY GENERATING PROCESS

Let $\tau_n = (V_n, E_n)$ be a tree topology with $n$ leaf nodes and $V_n, E_n$ are the sets of nodes and edges respectively. Here we only discuss the modeling of unrooted tree topologies. A pre-selected order (also called the taxa order) for the leaf nodes $\mathcal{X} = \{x_1, \ldots, x_N\}$ is assumed. We first give the definition of ordinal tree topologies.

**Definition 1** (Ordinal Tree Topology; Definition 1 in Xie & Zhang (2023)). *Let $\mathcal{X} = \{x_1, \ldots, x_N\}$ be a set of $N(N \geq 3)$ leaf nodes. Let $\tau_n = (V_n, E_n)$ be a tree topology with $n(n \leq N)$ leaf nodes in $\mathcal{X}$. We say $\tau_n$ is an ordinal tree topology of rank $n$, if its leaf nodes are the first $n$ elements of $\mathcal{X}$, i.e., $V_n \cap \mathcal{X} = \{x_1, \ldots, x_n\}$.*

The tree topology generating process is initialized by $\tau_3$, the unique ordinal tree topology of rank 3. In the $n$-th step ($n$ start from 3), assume we have an ordinal tree topology $\tau_n = (V_n, E_n)$ of rank $n$. To incorporate the leaf node $x_{n+1}$ into $\tau_n$, the following steps are taken:

1. A choice is made for an edge $e_n = (u, v) \in E_n$, which is then removed from $E_n$.
2. Add a new node $w$ and two additional edges, $(u, w)$ and $(w, v)$ to the tree topology $\tau_n$.
3. Add the next leaf node $x_{n+1}$ and an additional edge $(w, x_{n+1})$ to the tree topology $\tau_n$.

The above steps create an ordinal tree topology $\tau_{n+1}$ of rank $n + 1$. Repeating these steps for $n = 3, \ldots, N - 1$ leads to the eventual formation of the ordinal tree topology $\tau = \tau_N$ of rank $N$. The selected edges at each time step form a sequence $D = (e_3, \ldots, e_{N-1})$, which we call $D$ a decision sequence. Here we give two main theoretical results.

**Theorem 1.** *The generating process $g(\cdot) : D \mapsto \tau$ is a bijection between the set of decision sequences of length $N - 3$ and the set of ordinal tree topologies of rank $N$.*

**Theorem 2.** *The time complexity of the decomposition process induced by $g^{-1}(\cdot)$ is $O(N)$.*

The bijectiveness in Theorem 1 implies that we can model the distribution $Q(\tau)$ over tree topologies by modelling $Q(D)$ over decision sequences, i.e.,

$$Q(\tau) = Q(D) = \prod_{n=3}^{N-1} Q(e_n | e_{<n}), \tag{18}$$

where $e_{<n} = (e_3, \ldots, e_{n-1})$ and $e_{<3} = \emptyset$. The conditional distribution $Q(e_n | e_{<n})$, which describes the distribution of edge decision given all the decisions made previously, is called the edge decision distribution by us.

### B.2   GRAPH NEURAL NETWORKS FOR EDGE DECISION DISTRIBUTION

The edge decision distribution $Q(e_n | e_{<n})$ defines the probability of adding the leaf node $x_{n+1}$ to the edge $e_n$ of $\tau_n$, conditioned on all the ordinal tree topologies ($\tau_3, \ldots, \tau_n$) generated so far. To model $Q(e_n | e_{<n})$, ARTree employs the following four modules.

**Node embedding module**   At the $n$-th step of the generation process, ARTree relies on the node embedding module to assign node embeddings for the nodes of the current tree topology $\tau_n = (V_n, E_n)$. The embedding method follows Zhang (2023), which first assigns one-hot encoding for the leaf nodes:

$$[f_n(x_i)]_j = \delta_{ij}, \quad 1 \leq i \leq n, \quad 1 \leq j \leq N,$$

---

**Algorithm 2:** Two-pass algorithm for topological embeddings for internal nodes (Zhang, 2023)

---

**Input:** Tree topology $\tau_n = (V_n, E_n)$ of rank $n$, where $V_n = V_n^b \cup V_n^o$; Topological embeddings
for the leaf nodes $\{f_n(u)|u \in V_n^b\}$.
**Output:** Topological embeddings for the leaf nodes $\{f_n(u)|u \in V_n^o\}$
Initialized $c_u = 0, d_u = f_n(u)|u \in V_n^b$;
**for** $u$ *in the postorder traverse of* $\tau_n$ **do**
    **if** $u$ *is not the root node* **then**
        Compute

$$c_u = \frac{1}{|\mathcal{N}(u)| - \sum_{v \in \text{ch}(u)} c_v}, \quad d_u = \frac{\sum_{v \in \text{ch}(u)} d_v}{|\mathcal{N}(u)| - \sum_{v \in \text{ch}(u)} c_v}$$

        where $\mathcal{N}(u)$ is the neighborhood of $u$ and $\text{ch}(u)$ is the set of the children of $u$.
**end**
**for** $u$ *in the preorder traverse of* $\tau_n$ **do**
    **if** $u$ *is not the root node* **then**
        Let $f_n(u) = c_u f_n(\pi_u) + d_u$ where $\pi_u$ is the parent of $u$.
    **else**
        Let $f_n(u) = \frac{\sum_{v \in \text{ch}(u)} d_v}{|\mathcal{N}(u)| - \sum_{v \in \text{ch}(u)} c_v}$.
    **end**
**end**

---

where $\delta$ denotes the Kronecker delta function. We then obtain embeddings for the interior nodes by minimizing the Dirichlet energy, defined as

$$\ell(f_n, \tau_n) := \sum_{(u,v) \in E_n} ||f_n(u) - f_n(v)||^2.$$

This minimization process is achieved through the two-pass algorithm (Algorithm 2). Note that this process contains $(2n - 6)$ sub-iterations and each sub-iteration contains a linear combination over at most 3 vectors in $\mathbb{R}^N$. The time complexity of calculating the topological node embeddings is $O(Nn)$. Finally, a linear transformation is applied to all the node embeddings to obtain the initial node features in $\mathbb{R}^d$ for message passing. It should be highlighted that the embeddings for interior nodes may vary as the number of leaf nodes $n$, leading to the need for time guidance in the readout module.

**Message passing module** ARTree employs iterative message passing rounds to calculate the node features, capturing the topological information of $\tau_n$. The $l$-th message passing round is implemented by

$$m_n^l(u, v) = F_{\text{message}}^l(f_n^l(u), f_n^l(v)),$$
$$f_n^{l+1}(v) = F_{\text{updating}}^l\left(\{m_n^l(u, v); u \in \mathcal{N}(v)\}\right),$$

where $F_{\text{message}}^l$ and $F_{\text{updating}}^l$ are the message function and updating function in the $l$-th round, and $\mathcal{N}(v)$ is the neighborhood of the node $v$. The corresponding time-complexity is $O(nd^2)$ (noting that MLPs are applied to all the nodes) In particular, ARTree sets the number of message passing steps $L = 2$ and utilizes the edge convolution operator (Wang et al., 2018) for the design of $F_{\text{message}}^l$ and $F_{\text{updating}}^l$.

**Recurrent module** To efficiently incorporate the information of previously generated tree topologies into the edge decision distribution, ARTree uses a gated recurrent unit (GRU) (Cho et al., 2014) to form the hidden states of each node. Concretely, the recurrent module is implemented by

$$h_n(v) = \text{GRU}(h_{n-1}(v), f_n^L(v)),$$

where $h_n(v)$ is the hidden state of $v$ at the $n$-th step in the generating process For the newly added nodes, their hidden states are initialized to zeros. This module is mainly composed of MLPs on the node/edge features, whose time complexity is $O(nd^2)$.

**Readout module**   In the readout module, to form the edge decision distribution $Q(e_n|e_{<n})$, ARTree calculates the scalar edge feature $r_n(e) \in \mathbb{R}$ of $e = (u, v)$ using

$$p_n(e) = F_{\text{pooling}}\left(h_n(u) + b_n, h_n(v) + b_n\right),$$
$$r_n(e) = F_{\text{readout}}\left(p_n(e) + b_n\right),$$

where $b_n$ is the sinusoidal positional embedding of time step $n$ that is widely used in Transformers (Vaswani et al., 2017), $F_{\text{pooling}}$ is the pooling function implemented as 2-layer MLPs followed by an elementwise maximum operator, and $F_{\text{readout}}$ is the readout function implemented as 2-layer MLPs with a scalar output. This module is mainly composed of MLPs on the node/edge features, whose time complexity is $O(nd^2)$. The edge decision distribution is

$$Q(\cdot|e_{<n}) \sim \text{Discrete}\left(q_n\right), \quad q_n = \text{softmax}\left(\{r_n(e)\}_{e \in E_n}\right),$$

where $q_n \in \mathbb{R}^{|E_n|}$ is a probability vector.

Let $\phi$ be all the learnable parameters in GNNs. Then the ARTree based probability of a tree topology $\tau$ takes the form

$$Q_\phi(\tau) = Q_\phi(D) = \prod_{n=3}^{N-1} Q_\phi(e_n|e_{<n}),$$

The whole process of ARTree for generating a tree topology is summarized in Algorithm 3. An illustration of ARTree is in Figure 5.

### B.3   DISCUSSIONS ON THE COMPUTATIONAL COMPLEXITY OF ARTREE

During the above introduction of ARTree, we have given the time complexity of node embedding model $O(Nn)$ and that of the message passing module $O(nd^2)$, in each leaf node addition operation on subtree. Note that the leaf node addition operation should be repeated for $N = 3, \ldots, N$, which gives the overall time complexity of $O(N^3)$ and $O(N^2 d^2)$.

Note that the vectorized operations on tensors can be efficiently computed in PyTorch. We assume the $\alpha \in (0, 1)$ as the accelerated complexity order of the vectorized linear operations. We compute the accelerated time complexity as follows.

- For the node embedding module of ARTree, the cubic order of $N$ can be split into three aspects: (i) Iterating $N$ subtrees when autoregressively adding leaves; (ii) Iterating all $N$ internal nodes when computing the embeddings of a subtree; (iii) Summation and scalar multiplication of $N$-dimension vectors. Only (iii) can be accelerated with the vectorized operation, and (i) & (ii) always lead to two for-loops even if implemented in C++ or the fix-point algorithm. (Note that the number of fix-point iterations until convergence is $O(N)$.) This gives a complexity (with vectorization) of $O(N^{2+\alpha}) = O(N) \cdot O(N) \cdot O(N^\alpha)$.

- For the message passing module of ARTree, there is only one for-loop: Iterating $N$ subtrees when autoregressively adding leaves. Other computations in $N$ nodes and $d$-dimension features can be vectorized. Therefore, this gives an accelerated complexity of $O(N^{1+\alpha} d^{2\alpha}) = O(N) \cdot O(N^\alpha d^{2\alpha})$.

During introducing ARTreeFormer in Section 3.2, we have given the time complexity of node embedding model $O(nd + d^2)$ and that of the message passing module $O(d^2)$. Note that the leaf node addition operation should be repeated for $N = 3, \ldots, N$, which gives the overall time complexity of $O(N^2 + Nd^2)$ and $O(Nd^2)$. Regarding vectorization, all computations can be vectorized except for Iterating $N$ subtrees when autoregressively adding leaves. Therefore, the the complexity with vectorization is $O(N^{1+\alpha} d^\alpha + Nd^{2\alpha})$ and $O(Nd^{2\alpha})$.

## C   DETAILS OF VARIATIONAL BAYESIAN PHYLOGENETIC INFERENCE

By positing a tree topology variational distribution $Q_\phi(\tau)$ and a branch length variational distribution $Q_\psi(q|\tau)$ which is conditioned on tree topologies, the variational Bayesian phylogenetic inference (VBPI) (Zhang & Matsen IV, 2019) approximates the phylogenetic posterior $p(\tau, q|Y)$ in equation (2)

---

**Algorithm 3:** ARTree: an autoregressive model for phylogenetic tree topologies (Xie & Zhang, 2023)

---

**Input:** A set $\mathcal{X} = \{x_1, \ldots, x_N\}$ of leaf nodes.
**Output:** An ordinal tree topology $\tau$ of rank $N$; the ARTree probability $Q(\tau)$ of $\tau$.
$\tau_3 = (V_3, E_3) \leftarrow$ the unique ordinal tree topology of rank 3;
**for** $n = 3, \ldots, N-1$ **do**

    Let $f_n(u) = c_u f_n(\pi_u) + d_u$ where $\pi_u$ is the parent of $u$;
    Calculate the probability vector $q_n \in \mathbb{R}^{|E_n|}$ using the current GNN model;
    Sample an edge decision $e_n$ from Discrete $(q_n)$ and assume $e_n = (u, v)$;
    Create a new node $w$;
    $E_{n+1} \leftarrow (E_n \backslash \{e_n\}) \cup \{(u, w), (w, v), (w, x_{n+1})\}$;
    $V_{n+1} \leftarrow V_n \cup \{w, x_{n+1}\}$;
    $\tau_{n+1} \leftarrow (V_{n+1}, E_{n+1})$;

**end**

$\tau \leftarrow \tau_N$;
$Q(\tau) \leftarrow q_3(e_3) q_4(e_4) \cdots q_{N-1}(e_{N-1})$.

---

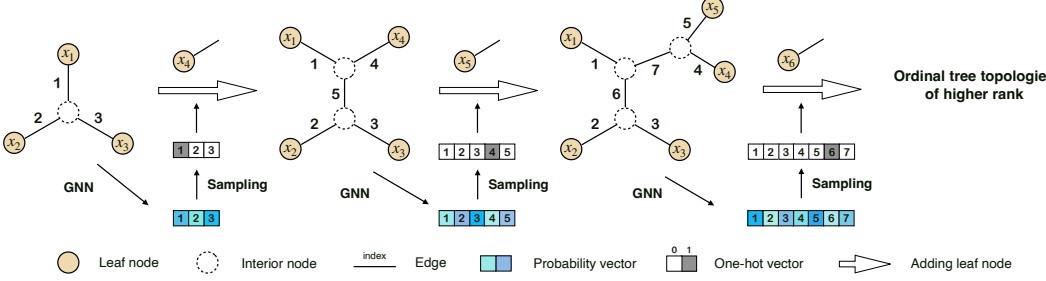

Figure 5: An illustration of ARTree starting from the star-shaped tree topology with 3 leaf nodes. This figure is from Xie & Zhang (2023).

with $Q_{\phi, \psi}(\tau, \boldsymbol{q}) = Q_\phi(\tau) Q_\psi(\boldsymbol{q}|\tau)$. To find the best approximation, VBPI maximizes the following multi-sample lower bound

$$L^K(\boldsymbol{\phi}, \boldsymbol{\psi}) = \mathbb{E}_{Q_{\phi, \psi}(\tau^{1:K}, \boldsymbol{q}^{1:K})} \log \left( \frac{1}{K} \sum_{i=1}^{K} \frac{p(\boldsymbol{Y}|\tau^i, \boldsymbol{q}^i) p(\tau^i, \boldsymbol{q}^i)}{Q_\phi(\tau^i) Q_\psi(\boldsymbol{q}^i|\tau^i)} \right).$$

where $Q_{\phi, \psi}(\tau^{1:K}, \boldsymbol{q}^{1:K}) = \prod_{i=1}^{K} Q_{\phi, \psi}(\tau^i, \boldsymbol{q}^i)$. Compared to the single-sample lower bound, the multi-sample lower bound enables efficient variance-reduced gradient estimators and encourages exploration over the vast and multimodal tree space. However, as a large $K$ may also reduce the signal-to-noise ratio and deteriorate the training of variational parameters (Rainforth et al., 2019), a moderate $K$ is suggested (Zhang & Matsen IV, 2024). In practice, the gradients of the multi-sample lower bound w.r.t the tree topology parameters $\phi$ and the branch length parameter $\psi$ can be estimated by the VIMCO/RWS estimator (Mnih & Rezende, 2016; Bornschein & Bengio, 2015) and the reparameterization trick (Kingma & Welling, 2014) respectively. Specifically, the gradient $\nabla_\phi L^K(\boldsymbol{\phi}, \boldsymbol{\psi})$ can be expressed as

$$\nabla_\phi L^K(\boldsymbol{\phi}, \boldsymbol{\psi}) = R_1 + R_2,$$

$$R_1 = \mathbb{E}_{Q_{\phi, \psi}(\tau^{1:K}, \boldsymbol{q}^{1:K})} \nabla_\phi \log \left( \frac{1}{K} \sum_{i=1}^{K} \frac{p(\boldsymbol{Y}|\tau^i, \boldsymbol{q}^i) p(\tau^i, \boldsymbol{q}^i)}{Q_\phi(\tau^i) Q_\psi(\boldsymbol{q}^i|\tau^i)} \right)$$

$$R_2 = \mathbb{E}_{Q_{\phi, \psi}(\tau^{1:K}, \boldsymbol{q}^{1:K})} \sum_{i=1}^{K} \log \left( \frac{1}{K} \sum_{i=1}^{K} \frac{p(\boldsymbol{Y}|\tau^i, \boldsymbol{q}^i) p(\tau^i, \boldsymbol{q}^i)}{Q_\phi(\tau^i) Q_\psi(\boldsymbol{q}^i|\tau^i)} \right) \nabla_\phi Q_{\phi, \psi}(\tau^i, \boldsymbol{q}^i).$$

VIMCO considers the following expression of $R_2$,

$$R_2 = \mathbb{E}_{Q_{\phi,\psi}(\tau^{1:K},\boldsymbol{q}^{1:K})} \sum_{i=1}^{K} \left\{ \log\left(\frac{1}{K}\sum_{i=1}^{K}\frac{p(\boldsymbol{Y}|\tau^i,\boldsymbol{q}^i)p(\tau^i,\boldsymbol{q}^i)}{Q_{\phi}(\tau^i)Q_{\psi}(\boldsymbol{q}^i|\tau^i)}\right) - \hat{f}_i \right\} \nabla_{\phi} Q_{\phi,\psi}(\tau^i,\boldsymbol{q}^i)$$

where $\hat{f}_i = \log\left(\frac{1}{K-1}\sum_{j\neq i}\frac{p(\boldsymbol{Y}|\tau^j,\boldsymbol{q}^j)p(\tau^j,\boldsymbol{q}^j)}{Q_{\phi}(\tau^j)Q_{\psi}(\boldsymbol{q}^j|\tau^j)}\right)$ is a control variate.

The tree topology model $Q_{\phi}(\tau)$ can be parametrized by ARTree, which enjoys unconfined support over the tree topology space. In addition to ARTree, subsplit Bayesian networks (SBNs) have long been the common choice for $Q_{\phi}(\tau)$. In SBNs, a subset $C$ of the leaf nodes is called a clade, and an ordered pair of two clades $(C_1, C_2)$ is called a subsplit of $C$ if $C_1 \cup C_2 = C$. For each internal node on a tree topology $\tau$, it corresponds to a subsplit $s$ determined by the descendant leaf nodes of its children. The SBNs are then parametrized by the probabilities of the root subsplit $\{p_{s_1}; s_1 \in \mathbb{S}_r\}$ and the probabilities of the child-parent subsplit pairs $\{p_{s|t};\ s|t \in \mathbb{S}_{\text{ch}|\text{pa}}\}$. For an unrooted tree topology $\tau = (V, E)$, its SBN based probability is

$$Q_{\text{sbn}}(\tau) = p_{s_r} \prod_{u \in V^o; u \neq r} p_{s_u|s_{\pi_u}},$$

where $V^o$ is the set of internal nodes, $r$ is the root node, $\pi_u$ are the parents of $u$, and $s_u$ is the subsplit assignment of the node $u$. As the size of $\mathbb{S}_r$ and $\mathbb{S}_{\text{ch}|\text{pa}}$ explodes combinatorially as the number of taxa increases, SBNs rely on subsplit support estimation for a tractable parameterization. **The subsplit support estimation can be difficult when the phylogenetic posterior is diffuse, and makes the support of SBNs cannot span the entire tree topology space.** We refer the readers to Zhang & Matsen IV (2018) and Zhang & Matsen IV (2019) for a detailed introduction to SBNs as well as their application to VBPI.

The branch length model $Q_{\psi}(\boldsymbol{q}|\tau)$ is often taken to be a diagonal lognormal distribution, which can be parametrized using the learnable topological features (Zhang, 2023) of $\tau$ as follows. This approach first assigns the topological node embeddings $\{f_u\}_{u \in V}$ to the nodes on $\tau$ (Algorithm 2) and then forms the node features $\{h_u\}_{u \in V}$ using message passing networks over $\tau$. Usually, these message passing networks take the edge convolutional operator (Wang et al., 2018). For each edge $e = (u, v)$ in $\tau$, one can obtain the edge features using $h_e = p(h_u, h_v)$ where $p$ is a permutation invariant function called the edge pooling. At last, the mean and standard deviation parameters for the diagonal lognormal distribution are given by

$$\mu(e, \tau) = \text{MLP}^{\mu}(h_e), \quad \sigma(e, \tau) = \text{MLP}^{\sigma}(h_e)$$

where $\text{MLP}^{\mu}$ and $\text{MLP}^{\sigma}$ are two multi-layer perceptrons (MLPs). In the VBPI experiment in Section 4.3, the collaborative branch length models for all SBN, ARTree, and ARTreeFormer are parametrized in this way.

## D  ADDITIONAL RESULTS ON TREE TOPOLOGY DENSITY ESTIMATION

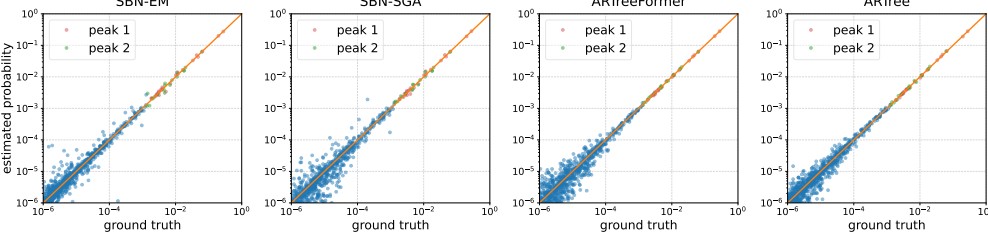

Figure 6: Performances of different methods for tree topology density estimation on DS1.

Figure 6 shows the performance of different methods on DS1. Both ARTree and ARTreeFormer provide more accurate probability estimates for the tree topologies on the two peaks of the posterior distribution, compared to SBN-EM and SBN-SGA. We see that ARTreeFormer can provide the same accurate probability estimates as ARTree, which proves the effectiveness of ARTreeFormer.

For ARTreeFormer, we also conducted an ablation study about the number of heads $h$ and the hidden dimension $d$ in the multi-head attention block (Table 4). In most cases, the KL divergence gets better as the number of heads increases. Increasing the embedding dimension $d$ may have a negative impact, partially due to the introduced difficulty in optimization and the overfitting problem.

Table 4: KL divergences ($\downarrow$) to the ground truth obtained by ARTreeFormer with different hyper-parameters on TDE.

| Hyper-parameters | $h = 2, d = 100$ | $h = 4, d = 100$ | $h = 4, d = 200$ | $h = 8, d = 200$ |
|---|---|---|---|---|
| DS1 | 0.0073 | 0.0067 | 0.0053 | 0.0045 |
| DS2 | 0.0105 | 0.0102 | 0.0109 | 0.0106 |
| DS3 | 0.0781 | 0.0777 | 0.0877 | 0.0948 |
| DS4 | 0.0318 | 0.0320 | 0.0445 | 0.0413 |

# E  ADDITIONAL RESULTS ON VARIATIONAL BAYESIAN PHYLOGENETIC INFERENCE

For ARTreeFormer, we conducted an ablation study about the number of heads $h$, the hidden dimension $d$, and the number of particles $K$ in the multi-sample lower bound. The results are reported in Table 5 and Table 6. In VBPI, the marginal estimate likelihood (MLL) is more sensitive to the branch length model; as we only improve the tree topology model, the MLL difference between different parameters is not evident. We observe that increasing the number of heads $h$ generally improves results, while a smaller hidden dimension $d$ can lead to missing modes. Concerning the number of particles $K$, a small $K$ can occasionally hinder mode discovery, whereas a moderate $K$ tends to perform well in practice.

Table 5: The marginal log-likelihood estimates obtained by ARTreeFormer with different hyper-parameters $h, d$ on VBPI. The number of particles is fixed as $K = 10$.

| Hyper-parameters | $h = 2, d = 50$ | $h = 2, d = 100$ | $h = 4, d = 100$ | $h = 4, d = 200$ | $h = 8, d = 200$ |
|---|---|---|---|---|---|
| DS1 | -7108.41(0.18) | -7108.41(0.15) | -7108.40(0.21) | -7108.42(0.18) | -7108.41(0.25) |
| DS2 | -26367.71(0.08) | -26367.71(0.08) | -26367.71(0.09) | -26367.71(0.07) | -26367.71(0.10) |
| DS3 | -33758.92(0.09) | -33735.10(0.08) | -33735.09(0.08) | -33735.10(0.08) | -33735.10(0.07) |
| DS4 | -13330.02(0.17) | -13332.43(0.25) | -13329.94(0.20) | -13329.94(0.21) | -13329.94(0.20) |

Table 6: The marginal log-likelihood estimates obtained by ARTreeFormer with different hyper-parameters $K$ on VBPI. The number of heads is fixed as $h = 4$ and the number of dimension is fixed as $d = 100$.

| Hyper-parameters | $K = 5$ | $K = 10$ | $K = 20$ |
|---|---|---|---|
| DS1 | -7108.42(0.17) | -7108.40(0.21) | -7108.41(0.14) |
| DS2 | -26367.70(0.09) | -26367.71(0.09) | -26367.71(0.07) |
| DS3 | -33751.34(0.08) | -33735.09(0.08) | -33735.09(0.09) |
| DS4 | -13332.51(0.21) | -13329.94(0.20) | -13329.95(0.17) |

To fully demonstrate the computational burden of ARTreeFormer compared to ARTree, we report the parameter size and memory usage of ARTreeFormer and ARTree for VBPI in Table 7. We see that ARTreeFormer has less memory consumption compared to ARTree, because ARTreeFormer only locally updates the node features, in analogy with the shorter sequence length in natural language modeling.

Table 8 compares the training time of SBN, ARTree, ARTreeFormer, and GeoPhy. Among these methods, SBN is the fastest because it only explores a fairly constrained subset of the tree topology space. The other two methods, ARTreeFormer and ARTree, are autoregressive models that explore the entire tree topology space. Although ARTreeFormer can cost more time than GeoPhy, it achieves much better approximation accuracy.

Table 7: The parameter size and memory usage of ARTreeFormer and ARTree for VBPI.

| Data set | DS1 | DS2 | DS3 | DS4 | DS5 | DS6 | DS7 | DS8 |
|---|---|---|---|---|---|---|---|---|
| ARTree (learnable parameter size) | 194K | 195K | 197K | 199K | 203K | 203K | 207K | 209K |
| ARTreeFormer (learnable parameter size) | 238K | 239K | 240K | 241K | 243K | 243K | 245K | 246K |
| ARTree (memory) | 1143MB | 1395MB | 1376MB | 1680MB | 1817MB | 1698MB | 2070MB | 2148MB |
| ARTreeFormer (memory) | 643MB | 720MB | 862MB | 915MB | 1041MB | 1151MB | 1333MB | 1372MB |

Table 8: Training time (seconds) per passing 100 trees of four methods on VBPI. The experiments are run on a single 2.4GHz CPU.

| Method | ARTreeFormer | ARTree | SBN | GeoPhy |
|---|---|---|---|---|
| DS1 (27 leaf nodes) | 2.49 | 6.06 | 1.02 | 1.87 |
| DS8 (64 leaf nodes) | 7.27 | 26.77 | 2.05 | 2.90 |

