# OpenReview forum: "ARTreeFormer: A Faster Attention-based Autoregressive Model for Phylogenetic Inference"
_ICLR.cc/2025/Conference — Submitted to ICLR 2025_

### Official Review · Reviewer_2SL2 · 2024-10-31

**Soundness:** 3
**Presentation:** 3
**Contribution:** 3
**Rating:** 6
**Confidence:** 3

**Summary:**

This submission extends the existing ARTree model for inferring phylogenetic posterior distributions by using attention mechanisms to compute the node embeddings. The resulting algorithm, ARTreeFormer, furthermore employs a constrained message-passage scheme which in total improves the run times of the algorithm in contrast to ARTree. The algorithm is evaluated on popular datasets commonly used in the VI-based phylogenetic inference literature.

**Strengths:**

The paper is well-written, and the proposed method tackles a problem which is of interest to the ML community and, importantly, reduces run times by significant factors. The use of attention mechanisms to tackle important problems is also aligned with the interests of the ICLR community. I think the submission is well-suited for publication at ICLR and that others will find it interesting, however I have some (actionable) concerns, given below.

**Weaknesses:**

Apart from the impressive improvements in run times, the results in terms of posterior accuracy (Table 1) and ML estimates (Table 2) are modest. For the ML estimates, this is expected as the scores are highly saturated, why most of the other works referenced in this submission instead value the size of the standard deviation (smaller is better). W.r.t. std, ARTReeFormer never outperforms the other VBPI methods. Regarding this I have one question and one comment:

* Question: Why do you think the std is larger for ARTreeFormer than the other VBPI methods? It would be nice to see a short discussion about this. The run times are already very good, so there is nothing to loose from transparently discussing this limitation of ARTreeFormer.
* Comment: There are other VBPI models available, such as [1,2,3,4], all of which (I believe) use pre-computed candidate trees. Notably [1] and [2] report better ML scores than those reported in Table 2 in this submission. See for instance DS8 in Table 2 in [1]. As such, the following statement in the current submission does not seem to hold: "Compared to other VBPI variants, the superiority of ARTreeFormer is mainly reflected by its unconfined support[...]". (This point weights down my soundness score).

Apart from my comment above, I think [1] and [2] should be referenced in the submission to demonstrate the growing interest in VI-based phylogenetic inference in the ML community. (This point weights down my presentation score).

In [5] they show that more importance samples during training can harm the inference of the variational approximation. In Eq. 3 in the submission it states that the models are trained using the importance weighted ELBO. How does the choice of $K$ affect the quality of your inferred approximations? Can you say something about the expected differences in scores for say $K=1$ or $K=100$?

If my concerns above are handled/discussed, I am willing to raise my overall rating to an acceptance level.

[1] https://arxiv.org/abs/2310.00941

[2] https://arxiv.org/abs/2406.07083

[3] https://pmc.ncbi.nlm.nih.gov/articles/PMC9949155/pdf/nihpp-2302.08840v1.pdf

[4] https://proceedings.neurips.cc/paper/2020/hash/d96409bf894217686ba124d7356686c9-Abstract.html

[5] https://arxiv.org/abs/1802.04537

**Questions:**

I am aware that there are many papers now in ML conferences on VI-based phylogenetics. Something that I am missing in these papers is a strong motivation of why better phylogenetic posterior inference algorithms are needed? As I have stated above, most of the results on the considered experiments appear saturated, and so it would really increase the significance of these papers if there were motivations that capture the necessity of these models and in which sense biologists would be interested in them. Do you think it is possible to provide such a motivation, potentially with references to works were similar types of models are used to draw biological conclusions?

Note that my question above is not a critique of your submission. It would however strengthen the paper, and potentially make it impactful outside of the ML community, if such a motivation was to be provided.

---

> ### Author Response · Authors · 2024-11-23
> **Official comment by authors**
>
> Thank you for your careful review and constructive feedback. Here are our responses to your concerns.
>
> **(W1)** Question: Why do you think the std is larger for ARTreeFormer than the other VBPI methods? (...)
>
> **Response**:
> The standard deviations of ARTreeFormer are larger because ARTreeFormer provides a slightly less accurate approximation than ARTree.
> However, the correct MLL means of ARTreeFormer suggests it works fairly well in terms of mode coverage.
> We have clarified this in our revision.
>
> **(W2)** Comment: There are other VBPI models available, such as [1,2,3,4], all of which (I believe) use pre-computed candidate trees. (...)
>
> **Response**:
> Thank you for providing these related works.
> We have cited VBPI-mixture [1][2] and discuss the potential improvement by employing mixtures in VBPI in Section 4.3.
> Note that although the above methods can provide better ML scores, they all require pre-computed candidate trees (which can be difficult to obtain) and hence would suffer from restricted support of tree topologies, especially when the posterior is diffuse [6].
> On the other hand, ARTree and ARTreeFormer enjoy unconfined support due to the autoregressive models for tree topologies that naturally cover the entire tree space.
> This is why we say compared to SBN-based VBPI variants, the superiority of ARTreeFormer is mainly reflected by its unconfined support.
> We'd like to clarify that **what we want to emphasize in this sentence is the methodological improvement instead of the accuracy gain**.
> We will modify this sentence to be more concise about ARTreeFormer's advantages over other baseline methods in our revision.
>
>
> **(W3)** Apart from my comment above, I think [1] and [2] should be referenced (...)
>
> **Response**:
> Thanks for providing these related works. We have cited [1,2] in our revision.
>
> **(W4)** In [5] they show that more importance samples during training can harm the inference of the variational approximation. (...)
>
> **Response**:
> Thanks for your question! We expect a moderate K to perform well in practice. That is because a small K would have limited exploration efficiency and hence hinder mode discovery, while a large K may lead to lower signal-to-noise ratio and hence deteriorate the training of the variational approximations as shown in [5].
> It is hard to tell exactly the differences between $K=1$ and $K=100$.
> Roughly speaking, $K=1$ may lead to a variational approximation that gets stuck at limited modes and $K=100$ may have good mode coverage but worse overall approximation quality.
>
> **(Q1)** I am aware that there are many papers now in ML conferences on VI-based phylogenetics. (...)
>
> **Response**:
> Thanks for your question! The main motivation is to develop efficient Bayesian phylogenetic inference algorithms that are capable of delivering accurate posterior estimates in a timely manner.
> The commonly used MCMC tends to be slow and often requires long runs to generate high quality samples. As an alternative, variational inference can provide comparable posterior estimation to MCMC, while requiring less computation due to a more efficient tree exploration mechanism and its optimization nature. Therefore, these VI-based phylogenetic inference algorithms have the potential to scale up Bayesian phylogenetic inference. As phylogenetic inference can provide extremely helpful information when making important public health decisions (especially during pandemics), it would be invaluable (with the potential to save thousands of lives) if significant computational efficiency improvement of Bayesian phylogenetic inference methods can be made in practice.
> For example, VI methods have been used for rapid analysis of pandemic-scale data (e.g.,  SARS-CoV-2 genomes) to provide accurate estimates of epidemiologically relevant quantities that are verifiable via alternative sources of public health data [7]. We have added some related discussion to the conclusion section in our revision.
>
> [6] Cheng Zhang and Frederick A Matsen IV. A variational approach to Bayesian phylogenetic inference.
> Journal of Machine Learning Research, 25(145):1–56, 2024.\
> [7] Caleb Ki and Jonathan Terhorst. Variational phylodynamic inference using pandemic-scale
> data. Mol. Biol. Evol., July 2022.

---

### Official Review · Reviewer_FrST · 2024-11-01

**Soundness:** 2
**Presentation:** 3
**Contribution:** 2
**Rating:** 5
**Confidence:** 4

**Summary:**

The ARTreeFormer model introduces an innovative attention-based approach to accelerate phylogenetic inference, addressing computational efficiency challenges in large-scale phylogenetic studies. Through recursive node embedding and local message-passing mechanisms, ARTreeFormer significantly reduces computational time while maintaining high performance across various tasks. Key improvements include the effective use of Transformer-based attention to prioritize meaningful interactions and the capability for parallel processing, enhancing its suitability for handling complex tree structures.

**Strengths:**

1. The field has made significant contributions and solved important challenges in phylogenetic inference: The introduction of the ARTreeFormer model has important academic and practical significance. Phylogenetic inference is extremely valuable in biological and ecological research, but existing methods face efficiency bottlenecks when processing large-scale data. Through innovative design, ARTreeFormer accelerates the inference process without significantly sacrificing accuracy, filling the gap in efficient phylogenetic inference models and promoting research and practical applications in the field.
2. Significantly improves efficiency and is suitable for large-scale data analysis: ARTreeFormer effectively reduces the computational time of the model through a recursive node embedding and local message passing strategy using an attention mechanism. This design allows the model to quickly generate and update tree topologies, which is especially advantageous when dealing with large numbers of nodes and complex tree structures. Experimental results show that ARTreeFormer is significantly superior to traditional models in terms of running time, making it highly competitive in processing large-scale phylogenetic data and promising for use in diverse biological data that requires efficient inference.

**Weaknesses:**

1. Limitations of performance improvement: Although ARTreeFormer shows significant improvement in computational efficiency, the improvement in inference accuracy is limited. In the tasks of maximum simplicity and tree topology density estimation, ARTreeFormer is only slightly better than existing models such as SBN-SGA and ARTree on some datasets (see the KL divergence results in Table 1). This result shows that the main advantage of ARTreeFormer is concentrated on computational speed, and the accuracy bottleneck has not yet been completely broken through. If the accuracy can be further improved, especially in terms of generalization ability and performance in complex scenarios, the model value and applicability will be greatly improved, making it more competitive for practical applications.
2. Model complexity increases and implementation difficulty: ARTreeFormer introduces a multi-layer attention mechanism and a local message passing module, which not only accelerates the calculation but also makes the model structure more complex. In “Section 3.2 Attention-Based Edge Decision Distribution”, the use of multi-head attention mechanisms increases the implementation complexity of the model and may place higher requirements on technical implementation. This complexity may limit the promotion and application of the model, especially in scenarios where resources are limited or efficient parallelization is difficult. Therefore, a better balance between structural optimization and implementation difficulty will help to improve the potential for widespread application of the model.
3. Lack of generalization verification on diverse datasets: The experimental evaluation mainly focused on eight specific phylogenetic datasets (“DS1-8”), which, although representative, have limited coverage. The applicability and robustness of the model to other types of biological data (such as viral evolutionary data or ecosystem data) have not been fully considered in the article, making it difficult to verify its generalization ability in diverse phylogenetic scenarios. Extending the experiment to a wider range of datasets would help to more comprehensively evaluate the applicability of ARTreeFormer, thereby enhancing the credibility of the model's application in the fields of bioinformatics and ecology.
4. Potential information loss in local message passing: Although the local message passing mechanism significantly reduces the computational burden, it may lead to potential global information loss. As mentioned in “Section 3.2 Local Message Passing Module”, only the neighborhood of the added node is updated, and the potential information loss of this mechanism in long-distance dependencies or complex tree structures is not discussed. Although the local update strategy improves efficiency, there may be a trade-off in terms of information integrity, which may affect model performance, especially in cases where node associations are complex or global consistency requirements are high. In the future, finding a better balance between efficient local updates and global information access will help improve the overall performance of the model.
5. Theoretical improvement: The depth of the explanation is insufficient. The article fails to provide sufficient theoretical support for several innovative points. For example, “Equation 12” only describes the calculation method of the local Dirichlet energy, without discussing in detail the theoretical basis of this energy minimization strategy and its relative advantages over global minimization. In addition, the specific optimization strategy of the local message passing mechanism is only briefly described. A more in-depth theoretical analysis would help to enhance the persuasiveness of the model design, so that readers can better understand its applicability and rationality in complex inference tasks.
6. Limitations of parallel processing in practical applications: ARTreeFormer supports the parallelization of node embedding and message passing on different tree topologies, which improves processing speed. However, “Figure 2 (right)” shows the significant acceleration effect of parallelization compared to non-parallelization, but does not discuss the feasibility in resource-constrained or single-machine environments. Although parallelization significantly improves running efficiency, its feasibility and practicality in environments with limited hardware resources still needs to be further verified. Optimizing model efficiency in single-machine or low-resource environments will be more conducive to the popularization of this model in practical applications.

**Questions:**

1. Extensiveness of experimental data sets and the problem of updating: Current experiments focus on the DS1 to DS8 data sets, which are relatively old and have a limited sequence length, making it difficult to fully reflect the diversity and complexity of modern biological data. It is recommended that tests be conducted on updated and diverse data sets (such as longer sequence data or more complex phylogenetic tree structures) to more comprehensively evaluate the robustness and generalization ability of ARTreeFormer.
2. How to use Transformer to increase the interpretability of the model and open the black box model: The Transformer's attention mechanism is introduced in ARTreeFormer, which, despite enhancing performance and efficiency, still has limited interpretability. The attention weights of the Transformer provide a potential interpretable pathway. For example, the key nodes and paths that the model focuses on when generating tree topology can be revealed by analyzing the attention distribution. However, the current model lacks a systematic interpretative mechanism. Future research can further explore how to use the attention weights to reveal the decision-making process of ARTreeFormer, helping to open the black box and make it more transparent and understandable in the task of biological evolution inference.

---

> ### Author Response · Authors · 2024-11-23
> **Official comment by authors (1/2)**
>
> Thank you for your careful review and constructive feedback. Here are our responses to your concerns.
>
> **(W1)** Limitations of performance improvement
>
> **Response**: We would like to clarify that there is no "accuracy bottleneck"; SBN, ARTree, and ARTreeFormer are all the SOTA methods for VBPI, and work much better than other models (see Table 3).
> The bottleneck of SBN is that it requires support estimation and hence suffers from a confined support and the bottleneck of ARTree is the large computation cost. These two bottlenecks are both overcome by ARTreeFormer.
> We admit there is an accuracy loss of ARTreeFormer compared to ARTree, but this is negligible for practical applications.
>
> *Table: KL divergence difference between different methods on TDE*
> | | DS1 | DS2 | DS3 | DS4 | DS5 | DS6 | DS7 | DS8 |
> |----|----|----|----|----|----|----|----|----|
> |ARTreeFormer - ARTree |0.0022|0.0005|0.0229|0.0021|0.0415|0.0118|0.0080|-0.0074|
> |SBN-EM-$\alpha$ - ARTree|0.0085|0.0031|0.0334|0.0341|0.1952|0.0426|0.0208|0.0495|
>
> **(W2)** Model complexity increases and implementation difficulty
>
> **Response**: We have provided the Python implementation of ARTreeFormer which can be readily used for further applications.
> The message passing steps, attention mechanisms, etc, are common techniques in graph learning methods.
> The model architecture of ARTreeFormer is not complex -- instead, we make a careful design that enables vectorized computation. Benefiting from this, the actual implementation of ARTreeFormer is highly organized and batched, which also greatly improves the computational speed and does not show additional complexity than other graph models.
> See Figure 2 for a detailed runtime comparison of different modules.
> In contrast, it is the inefficient design (although easy to describe) of ARTree, e.g., the multiple and irreducible loops (repetitive dirichlet energy minimization for topological node embeddings), that causes the highly inefficient computations. Therefore, a more efficient and faster design is very necessary.
> See a more detailed discussion on page 6 (from line 301 to line 323).
>
>
> **(W3)** Lack of generalization verification on diverse datasets
>
> **Response**:
> We would like to clarify that DS1-8 are commonly used benchmarks in the field of Bayesian phylogenetic inference [1,2,3,4,5].
> This is because DS1-8 covers all sorts of duplicated sub-structures and multimodal posteriors, as shown in [6].
> Therefore, we can expect ARTreeFormer to work well for other data sets if it works for DS1-8.
>
> [1] Zhang, C. and Matsen IV, F. A. "Variational Bayesian phylogenetic inference." International Conference on Learning Representations (2019).\
> [2] Koptagel, Hazal, et al. "Vaiphy: a variational inference based algorithm for phylogeny." Advances in Neural Information Processing Systems 35 (2022): 14758-14770.\
> [3] Mimori, Takahiro, and Michiaki Hamada. "GeoPhy: differentiable phylogenetic inference via geometric gradients of tree topologies." Advances in Neural Information Processing Systems 36 (2023).\
> [4] Xie, T. and Zhang, C. "ARTree: A deep autoregressive model for phylogenetic inference." Advances in Neural Information Processing Systems 36 (2023).\
> [5] Zhou, Mingyang, et al. "PhyloGFN: Phylogenetic inference with generative flow networks." arXiv preprint arXiv:2310.08774 (2023).\
> [6] Chris Whidden and Frederick A Matsen IV. Quantifying MCMC exploration of phylogenetic tree space. Syst. Biol., 64(3):472–491, May 2015.
>
>
> **(W4)** Potential information loss in local message passing
>
> **Response**:
> To inspect which component contributes to the reduced performance of ARTreeFormer, we conducted an ablation study. We see that the strong representation power of topological node embedding is the most crucial part of a model's performance. Importantly, **the attention-based local message passing improves both accuracy and efficiency**.
>
> *Table: KL divergence obtained by various model architectures on the TDE task.*
> | Model | DS1 | DS2 | DS3 | DS4 |
> | --- | --- | --- | --- | --- |
> | Topological Node Embedding + Global GNN ( = ARTree) | 0.0045 | 0.0097 | 0.0548 | 0.0299 |
> | Topological Node Embedding + Local Attention | 0.0089 | 0.0102 | 0.0441 | 0.0276 |
> | Learnable Node Embedding + Global GNN | 0.0044 | 0.0123 | 0.0916 | 0.0346 |
> | Learnable Node Embedding + Local Attention (= ARTreeFormer)| 0.0067 | 0.0102 | 0.0777 | 0.0320 |
>
> **(W5)** Theoretical improvement
>
> **Response**:
> The main contribution of this paper lies in the model architecture improvement and the superior acceleration results.
> Exploring the in-depth theoretical analysis is an interesting topic and we leave this for future work.

---

> > ### Author Response · Authors · 2024-11-23
> > **Official comment by authors (2/2)**
> >
> > **(W6)** Limitations of parallel processing in practical applications
> >
> > **Response**:
> > Thanks for this question. We assume that there are fairly sufficient machines for training models, in this prosperous era of AI.
> > More and more large models come into being, e.g., GPT-4, stable diffusion, which can require hundreds of GPUs for training.
> > Moreover, 8 CPU cores are enough for training ARTreeFormer (just on your laptop)!
> >
> > **(Q1)** Extensiveness of experimental data sets and the problem of updating
> >
> > **Response**:
> > We'd like to clarify DS1-8 are not old datasets, but are good representative datasets for various posterior patterns. Please check our response to W3 for detailed discussions.
> >
> > **(Q2)** How to use Transformer to increase the interpretability of the model and open the black box model
> >
> > **Response**:
> > Thanks for this constructive suggestion! We admit that exploiting the interpretation possibility of Transformer can be a very important direction.
> > We plan to do this in the future.

---

> > > ### Comment · Reviewer_FrST · 2024-11-27
> > > **Thank you for your detailed and thoughtful responses to my comments. However, after carefully considering the points raised by Reviewer 3iPi, I find their arguments compelling and aligned with my initial concerns. As such, I will maintain my current score.**
> > >
> > > Thank you for your detailed and thoughtful responses to my comments. However, after carefully considering the points raised by Reviewer 3iPi, I find their arguments compelling and aligned with my initial concerns. As such, I will maintain my current score.

---

### Official Review · Reviewer_3oNL · 2024-11-04

**Soundness:** 3
**Presentation:** 3
**Contribution:** 2
**Rating:** 5
**Confidence:** 4

**Summary:**

The paper proposes ARTreeFormer, which aims to improve the computational efficiency in phylogenetic inference tasks by improving the time complexity of the global multi-head attention mechanism (MHA). The motivation of this paper focuses on solving the computational efficiency problems caused by the minimisation of Dirichlet energy and message passing to all nodes. The innovation of the paper lies in the use of the Transformer architecture to achieve faster ancestor sampling and probability evaluation than the baseline ARTree with the improvement of reducing complexity from O(N3) to O(N2). The experimental part verifies the time efficiency and performance of the model and demonstrates its performance in specific benchmark tests.

**Strengths:**

* (S1) ARTreeFormer achieves a better computational complexity with the recurrent node embedding module using local operations in comparison to the traditional multi-head attention mechanism, which is innovative in terms of computational efficiency. It can be supported by comparison experiments with the improvement of the performance and efficiency on some specific tasks.

* (S2) The overall presentation is well-originaized and easy to follow. It is clear that the authors provide step-by-step designs to improve the attention mechanism for better performance and efficiency.

**Weaknesses:**

* (W1) This paper emphasizes the time complexity of improving the MHA module but without showing some ablation experiments that remove the MHA module, which might not be a fair verification to support the contribution of the MHA module. The novelty and motivation of this paper are somewhat unclear (refer to Q1).

* (W2) The proposed graph pooling function with the MHA and the message-passing process all learn local structures and node information, which might lack the aggregation of global information. The authors should verify whether these designs are reasonable for phylogenetic inference. Are there any intrinsic drawbacks or necessities of capturing long-distance dependencies or global topologies in these tasks?

* (W3) Since computational efficiency is an important contribution, the authors should clarify the discussion of the time complexity formula in line 213. But it is only briefly mentioned in the main text. I recommended the authors provide a more detailed derivation to show the complexity reduction from $O(N^3)$ to $O(N^2)$. Some detailed explanation of each module is unclear, especially the simplification process of the MHA.

**Questions:**

* (Q1) It seems that the motivation of this work is to address the computational efficiency of Dirichlet energy minimization and message passing to all nodes. Although the proposed module can achieve faster ancestor sampling and probability evaluation than ARTree without using the global MHA, the overall innovation still seems to be insufficient and needs clarification.

* (Q2) The practical value of the improvement results is unclear. The paper improves time efficiency but does not demonstrate the significance of this improvement in practical applications, nor does it provide a specific dataset or scenario to verify the actual necessity of such efficiency improvements in this field. The reduction of complexity from $O(N^3)$ to $O(N^2)$ is interesting, but it may be more suitable and important to improve the multi-head attention mechanism itself (e.g., using linear attentions and Mamba variants).

---

> ### Author Response · Authors · 2024-11-23
> **Official comment by authors**
>
> Thank you for your careful review and constructive feedback. Here are our responses to your concerns.
>
> **(W1)** This paper emphasizes the time complexity of improving the MHA module but without showing some ablation experiments that remove the MHA module (...)
>
> **Response**:
> Thanks for this constructive question. To inspect which component contributes to the reduced performance of ARTreeFormer, we conducted an ablation study. We see that whatever the node embedding module is, the attention-based local message passing always improves both accuracy (the following table) and efficiency (Figure 2).
>
> *Table: KL divergence obtained by various model architectures on the TDE task.*
> | Model | DS1 | DS2 | DS3 | DS4 |
> | --- | --- | --- | --- | --- |
> | Topological Node Embedding + Global GNN ( = ARTree) | 0.0045 | 0.0097 | 0.0548 | 0.0299 |
> | Topological Node Embedding + Local Attention | 0.0089 | 0.0102 | 0.0441 | 0.0276 |
> | Learnable Node Embedding + Global GNN | 0.0044 | 0.0123 | 0.0916 | 0.0346 |
> | Learnable Node Embedding + Local Attention (= ARTreeFormer)| 0.0067 | 0.0102 | 0.0777 | 0.0320 |
>
> **(W2)** The proposed graph pooling function with the MHA and the message-passing process all learn local structures and node information, which might lack the aggregation of global information.
>
> **Response**:
> We want to clarify that ARTreeFormer does not discard global information. First, in the node embedding module, MHA extracts all the node features to form the features of the new nodes. Second, we considered a global representation vector $r_n$ in forming edge decision distributions.
> ARTreeFormer is local only in the attention-based message passing module; however, the local attention can be more powerful than the global GNN, as suggested by the above table.
>
> **(W3)** Since computational efficiency is an important contribution, the authors should clarify the discussion of the time complexity formula in line 213.
>
> **Response**:
> Thanks for this suggestion. The following table compares the computational complexity of the two major modules in ARTree/ARTreeFormer, where $\alpha\in(0,1)$ refers to the accelerated complexity order due to vectorized operations in PyTorch.
>
> We see that $\alpha$ can be small in practice (i.e., fast computation of batched tensors in PyTorch). Even in such a case, the complexity of ARTree's node embedding module is still higher than $O(N^2)$, while those of other modules are reduced to approximately equal to or less than $O(N)$. This validates the observation that the topological node embedding dominates the computation time. We have added a detailed computation and explanation on this in our revision.
>
> |Module|ARTree|ARTreeFormer|
> |---|---|---|
> |Node embedding module|$O(N^3)$|$O(N^2d+Nd^2)$|
> |Message passing module|$O(N^2d^2)$|$O(Nd^2)$|
> |Node embedding module (vectorization)|$O(N^{2+\alpha})$|$O(N^{1+\alpha}d^{\alpha}+Nd^{2\alpha})$|
> |Message passing module (vectorization)|$O(N^{1+\alpha}d^{2\alpha})$|$O(Nd^{2\alpha})$|
>
> **(Q1)** It seems that the motivation of this work (...)
>
> **Response**:
> Thanks for this question. We have clarified our innovation in the last paragraph in Section 3, and put it here for your convenience.
> - First, the learnable node embedding based on the attention mechanism is novel and overcomes the non-vectorizable bottleneck of ARTree.
> - Second, we incorporate message passing and local Dirichlet energy minimization within the neighborhood structure, tailored for phylogenetic trees.
> - Third, adapting graph techniques to phylogenetic trees is not straightforward and requires careful design, and we are the first to show that this simplified attention-based architecture exhibits strong approximation capacity with considerably reduced computational cost.
>
> **(Q2)** The practical value of the improvement results is unclear. (...)
>
> **Response**:
> As we claimed in our paper, ARTree suffers from extremely large computational costs. For example, **it takes 12 days for training ARTree on DS8** (just multiply the training time in Figure 4 (right) by 40000.)
> Given that timely Bayesian phylogenetic analysis is crucial for making important public health decisions (especially during pandemics), it would be invaluable (with the potential to save thousands of lives) if significant computational efficiency improvement of Bayesian phylogenetic inference methods can be made in practice.
>
> We have also tried other types of attention mechanisms (e.g., linear attention) and report the results in the following table.
> We find that linear attention works fairly well but is inferior to softmax attention.
>
> *Table: KL divergence obtained by different attention types on the TDE task.*
> | Attention type \ Dataset | DS1 | DS2 | DS3 | DS4 |
> | --- | --- | --- | --- | --- |
> |softmax |0.0067|0.0102|0.0777|0.0320|
> |linear |0.0078|0.0108|0.0884|0.0406|

---

> > ### Comment · Reviewer_3oNL · 2024-12-03
> > **Feedback to Authors' Rebuttal**
> >
> > Thanks for the detailed response, and sorry for the late reply. Overall, I believe the authors have largely tackled my concerns. However, after viewing comments and discussions from other reviewers, I am not sure whether this manuscript is qualified enough to be accepted. Therefore, I lowered my confidence to 4 and kept my original score. I encourage the authors to further polish the revised manuscript with concerns I and other reviewers have raised.

---

### Official Review · Reviewer_3iPi · 2024-11-04

**Soundness:** 1
**Presentation:** 2
**Contribution:** 2
**Rating:** 3
**Confidence:** 4

**Summary:**

This paper introduces ARTreeFormer, a novel probabilistic model over tree topologies. ARTreeFormer is a variant of the previously proposed ARTree model, with the advantage that ARTreeFormer is faster while showing comparable accuracy. The speed improvement is achieved by replacing the topological node embedding module and the GNN edge-decision layer with (1) attention-based recurrent node embeddings and edge-decisions, (2) a local Dirichlet energy minimization procedure, and (3) a local message passing scheme which only updates the node embeddings in the neighbourhood of the newly added node. On three standard benchmarks (the large parsimony problem, TDE, and VBPI), the paper demonstrates that ARTreeFormer is ~3x faster on learning tasks while showing comparable accuracy to ARTree.

**Strengths:**

This paper demonstrates that topological node embeddings derived from Dirichlet energy minimization are not needed to achieve near-SoTA results. Instead, learnt attention-based recurrent node embeddings can provide similar performance (if not slightly worse - see Table 1). The use of attention-based edge decisions and the local message passing scheme are also new ideas wrt ARTree which provide speedups to the model.

**Weaknesses:**

For a work whose main contribution is a more computationally efficient model (in the author's words, line 338: "It should be emphasized that we mainly pay attention to the computational efficiency improvement of ARTreeFormer and only expect it to attain similar accuracy with baseline methods"), it is surprising that there is no analysis whatsoever of the computational complexity of ARTree and ARTreeFormer. My expectation coming into the paper was that (1) the paper would discuss the computational complexity of each main step of the ARTree model using Big-O/Theta/Omega notation, deriving the runtime as a function of key quantities such as the number of nodes in the tree ($n$), the node feature dimension ($d$), etc., and discussing the degree of vectorization/parallelism achievable in each step, (2) the paper would then go on to profile each step of ARTree to confirm that the theoretical runtimes of these step align with the empirical runtimes, thereby establishing the bottlenecks, and (3) the alternative ARTreeFormer model would be proposed and its runtime again analyzed theoretically and empirically. The paper does not attempt to perform any computational complexity analysis of this kind (with the exception of the comments on line 213 and line 266). This leaves me wondering which are the real computational bottlenecks and why. Or - are they the result of inefficient implementation?

In particular, the paper claims that the node embedding module is a big computational bottleneck: "As N increases, the total run time of ARTree grows rapidly and the node embedding module dominates the total time (≈ 65%), which makes ARTree prohibitive when the number of leaf nodes is large. The reason behind this is that compared to other modules, the node embedding module can not be easily vectorized and parallelized w.r.t. different tree topologies and different nodes, resulting in great computational inefficiency.". I am skeptical about this claim. By looking at the code provided by the authors, as well as at the original ARTree code, the implementation of Dirichlet energy minimization used by the node embedding module seems to be a highly inefficient Python implementation. Here is what I found:

```
def node_embedding(self, tree, level):
    name_dict = {}
    j = level
    rel_pos = np.arange(max(4,2*level-4))
    for node in tree.traverse('postorder'):
        if node.is_leaf():
            node.c = 0
            node.d = self.leaf_features[node.name]
        else:
            child_c, child_d = 0., 0.
            for child in node.children:
                child_c += child.c
                child_d += child.d
            node.c = 1./(3. - child_c)
            node.d = node.c * child_d
            if node.name != '':
                rel_pos[node.name] = j
            node.name, j = j, j+1
        name_dict[node.name] = node

    node_features, node_idx_list, edge_index = [], [], []
    for node in tree.traverse('preorder'):
        neigh_idx_list = []
        if not node.is_root():
            node.d = node.c * node.up.d + node.d
            neigh_idx_list.append(node.up.name)

            if not node.is_leaf():
                neigh_idx_list.extend([child.name for child in node.children])
            else:
                neigh_idx_list.extend([-1, -1])
        else:
            neigh_idx_list.extend([child.name for child in node.children])

        edge_index.append(neigh_idx_list)
        node_features.append(node.d)
        node_idx_list.append(node.name)

    branch_idx_map = torch.sort(torch.tensor(node_idx_list,device=self.device).long(), dim=0, descending=False)[1]
    edge_index = torch.tensor(edge_index,device=self.device).long()

    return torch.index_select(torch.stack(node_features), 0, branch_idx_map), edge_index[branch_idx_map], torch.from_numpy(rel_pos).to(self.device), name_dict
```

Notice not only the use of for loops which are notoriously slow in Python, but also the ete3 tree object which is traversed as part of the algorithm. An efficient implementation of Dirichlet energy minimization, say in C++, would likely be a lot faster (say >=10x). If so, ARTree might catch up to the speed of ARTreeFormer while giving slightly better results (see Table 1). Dirichlet energy minimization being a bottleneck is also suspicious given that it is a linear-time algorithm ($O(nk)$ where $n$ is the number of leaves and $k$ is the embedding size).

There are other natural options beyond an efficient C++ (or numba, or cython) implementation to potentially speed up the Dirichlet energy minimization, for example:
- The linear system $Ax=b$ involved in Dirichlet energy minimization has very particular, sparse structure. For example, one can perform fixed point iteration via $f^{t+1}(u) = 1/N(u) \sum_{w \in N(u)} f^{t}(w)$ until convergence. This is a (sparse) matrix-vector multiply $f^{t+1} = A'f^t$ for a suitable matrix $A'$. One can use repeated squaring for fast convergence ($A'^{2t} = (A'^{t})^2$). This is essentially just computing the eigenvector of the matrix $A'$ with repeated squaring. This might be fast and it is naturally vectorizable (just batch all the $A'$ matrices from different trees together).
- Using low-dimensional embeddings: to each leaf node assign a fixed random embedding on the $n'$-dimensional simplex, with $n' < n$, then set the internal node embeddings via Dirichlet energy minimization. This should be faster than using $n$-dimensional embeddings. Random embeddings usually have good statistical properties. Furthermore, since these embeddings get projected into dimension $d$ when computing the node features with the GNN in ARTree (as mentioned in line 166), it would seem reasonable to take e.g. $n'=d$. I.e why not just start in dimension $d$ to begin with?

Another comment regarding runtime analysis: runtimes for (1) generating tree topologies vs (2) learning the model parameters may differ because training requires gradients while generating does not. This means that while the node embedding module represents 65% of the time for tree generation (Figure 2) it may be less of a bottleneck for model training. Without proper profiling, it is hard to tell what the computational bottlenecks really are and whether the paper is solving any real computational bottlenecks at all, or just implicitly overcoming inefficient implementations. It is important to recall that ARTreeFormer performs slightly worse than ARTree on several tasks (see e.g. Table 1 where ARTree beats ARTreeFormer on all but 1 dataset). Thus, the architectural changes proposed by ARTreeFormer should be well justified. If speeding up ARTree's Dirichlet energy minimization step as above makes ARTree just as fast as ARTreeFormer, the value of ARTreeFormer drops significantly.

Architecturally speaking, ARTree (Algorithm 3 / Figure 5) is simpler and more elegant that ARTreeFormer (Algorithm 1 / Figure 1), since ARTree simply uses the Dirichlet node embeddings (which are interpretable and have some nice theoretical properties [Zhang 2023, "Learnable Topological Features for Phylogenetic Inference via Graph Neural Networks"]) into a GNN to obtain edge decisions. ARTreeFormer, in contrast, uses an ad-hoc arquitecture consisting of attention-based learnt recurrent node embeddings, local Dirichlet edge minimization, and a local message passing scheme (each of which might explain the reduced performance of ARTreeFormer in some cases, as acknowledged by the paper and seen in Table 1 - without proper ablations, which are missing, it is hard to know which module causes performance to be lost). I generally found it hard to understand the ARTreeFormer model (Section 3.2). It suffices to compare the schematics for the two models (ARTree - Figure 5, ARTreeFormer - Figure 1) to notice the increase in architecture  complexity. In particular, note the presence of cycles in ARTreeFormer in Figure 1. There seems to be unnecessary complexity in ARTreeFormer, which is hard to justify without proper runtime analysis.

The paper is plagued with vague qualitative statements such as:
- abstract: "significantly improves the computation speed"
- line 69: "much more computationally efficient"
- line 177: "require much computational cost".
- line 184: "resulting in great computational inefficiency".
- line 307: "greatly improved computational efficiency".
- line 313: "runtime [...] are significantly reduced".
- line 323: "considerably reduced computation cost".
- etc.

Please provide quantitative claims instead of qualitative ones. For example, say "3x faster".

The words "vectorization" and "parallelization" seem to be used interchangeably throughout the paper. I believe in standard use, "parallelization" is used to describe the process of running code on several processes in parallel, while "vectorization" is used to describe speeding up code via use of vector operations (such as matrix-matrix products) which can be efficiently implemented on GPUs or CPUs (e.g. via SIMD). The terms "vectorization" and "parallelization" seem to be used interchangeably in the paper, which leads to confusion. For example, in Fig. 2 right the paper shows the "runtime of ARTreeFormer for generating 100 tree topologies with and without parallelization". Is this parallelization as in multiprocessing, or as in vectorization? (e.g. via batching the trees). Since the figure caption says "All tests are run on a single 2.4 GHz CPU" I believe this is vectorization via batching, not parallelization.

Overall, the lack of proper profiling and computational complexity analysis, the suspicous Dirichlet energy minimization code which seems highly inefficient, the increased complexity of the model architecture which makes it hard to understand compared to ARTree, all without an increase in accuracy (as seen in Table 1), lead me to believe this work requires a major revision before being ready for publication, so I recommend rejection with a score of 3.

I believe if pursued in the right direction, the ideas in this paper could lead to not just faster but also more elegant and performant probabilistic models for tree topologies. The idea of learnt node embeddings and use of attention instead of global message passing all are appealing.

**Questions:**

1. What implementation of Dirichlet energy minimization are you benchmarking? Is it the Python implementation I cited above?
2. As given by Big-O/Theta/Omega notation, what is the computational complexity of each step of ARTree and ARTreeFormer, as a function of key parameters such as the number of nodes ($n$), the feature dimension ($d$), etc.?
3. Does the empirical runtime of each step align with the theoretical runtimes from question 2? For example, if the Dirichlet energy minimization step has total runtime of $\Theta(n^3)$ for sampling one tree, does it align with the empirical runtime? (A back-of-the-envelope calculation used frequently is that in a performant CPU implementation, e.g. in C++, $10^8$ to $10^9$ operations will be executed per second).
4. How costly is it to compute the gradients of the model parameters, which are needed for learning? This is not revealed by Figure 2, which only considers generation.
5. Line 283: "with or without parallelization" - how was the code "parallelized"? Provide details.
6. line 946: "two peaks of the posterior distribution": just curious, how do you know there are "two peaks"? I am not familiar with this dataset.
7. Have you considered other simple options to speed up ARTree's Dirichlet energy minimization step? I already mentioned two above.

---

> ### Author Response · Authors · 2024-11-23
> **Official comment by authors (1/2)**
>
> Thanks for your careful review and helpful suggestions! Here are our responses to your concerns.
>
> **Time complexity analysis**
> Thanks for your constructive suggestion!
> The following table compares the computational complexity of the two major modules in ARTree/ARTreeFormer, where $\alpha\in(0,1)$ refers to the accelerated complexity order of linear (1-order) vectorized operations in PyTorch.
> The results are interpreted as follows.
> - For the node embedding module of ARTree, the cubic order of $N$ can be split into three aspects: (i) Iterating $N$ subtrees when autoregressively adding leaves; (ii) Iterating all $N$ internal nodes when computing the embeddings of a subtree; (iii) Summation and scalar multiplication of $N$-dimension vectors. We note that only (iii) can be accelerated with the vectorized operation, and **(i) & (ii) always lead to two for-loops even if implemented in C++ or the fix-point algorithm**. (Note that the number of fix-point iterations until convergence is $O(N)$.) This gives a complexity (with vectorization) of $O(N^{2+\alpha})=O(N)\cdot O(N)\cdot O(N^\alpha)$.
> - For the message passing module of ARTree, there is only one for-loop: Iterating $N$ subtrees when autoregressively adding leaves. Other computations in $N$ nodes and $d$-dimension features can be vectorized. Therefore, this gives an accelerated complexity of $O(N^{1+\alpha}d^{2\alpha})=O(N)\cdot O(N^\alpha d^{2\alpha})$.
> - For the node embedding module and message passing module in ARTreeFormer, all computations can be vectorized except for "Iterating $N$ subtrees when autoregressively adding leaves". Therefore, the complexity with vectorization is $O(N^{1+\alpha}d^{\alpha}+Nd^{2\alpha})$ and  $O(Nd^{2\alpha})$.
>
> We would like to emphasize that $\alpha$ can be small in practice (i.e., fast computation of batched tensors in PyTorch). Even in such a case, the complexity of ARTree's node embedding module is still higher than $O(N^2)$, while those of other modules are reduced to approximately equal to or less than $O(N)$. This validates the observation that the topological node embedding dominates the computation time.
>
> |Module|ARTree|ARTreeFormer|
> |---|---|---|
> |Node embedding module|$O(N^3)$|$O(N^2d+Nd^2)$|
> |Message passing module|$O(N^2d^2)$|$O(Nd^2)$|
> |Node embedding module (vectorization)|$O(N^{2+\alpha})$|$O(N^{1+\alpha}d^{\alpha}+Nd^{2\alpha})$|
> |Message passing module (vectorization)|$O(N^{1+\alpha}d^{2\alpha})$|$O(Nd^{2\alpha})$|
>
> **Topological embedding with fix-point algorithm**
> We thank the reviewer for the suggestion of using a fix-point algorithm. We try this method and report the time consumption in the following Table, where we see **there is no evident improvement of fix-point algorithm**.
> The reasons are analyzed as follows:
> - There are three for-loops in the topological node embedding within the two-pass algorithm of ARTree: (a) a traversal over the nodes on a tree; (b) a traversal over all the trees in a batch; (c) apply (a)(b) to each ordinal subtree as the tree autoregressively grows. All these (a)(b)(c) loops cannot be vectorized in the current implementation.
> - For the topological node embedding with fix-point algorithm, (b) is vectorized and (a)(c) are kept. Note that the (a) is replaced by an iteration process in the fix-point algorithm, and the number of iterations until convergence is in the same order as the number of taxa (because the information should be propagated from the leaf nodes with initial features to the internal nodes).
> Finally, as the corresponding linear system is not homogeneous, the square trick $(A')^2$ cannot be applied.
> - For the learnable topological embedding in ARTreeFormer, (a)(b) are vectorized and (c) is kept.
>
> *Table: Time consumption (seconds) of node embedding module in generating 100 trees. "n'=n" means setting embedding dimension $n'$ as the number of leaf nodes $n$ in a subtree.*
> |Method\Number of leaves |10|20|30|40|50|60|70|80|90|100|
> |---|---|---|---|---|---|---|---|---|---|---|
> |Topological node embedding with fix-point algorithm (ARTree's default setting, n'=n, tol=1e-5)|0.07|0.28|0.61 |1.12|1.92|3.07|4.46|6.38|8.78 |11.72|
> |Topological node embedding with fix-point algorithm ($n'=d=100$, tol=1e-5)|0.13|0.55|1.37|2.48|3.85|5.62|7.72|10.14|12.94|16.01|
> |Topological node embedding with fix-point algorithm ($n'=d=50$, tol=1e-5)|0.1|0.33|0.72|1.25|1.97|2.89|4.|5.2|6.61|8.19|
> |Topological node embedding with two-pass algorithm|0.18|0.78| 1.97|3.33|5.31|7.52|10.58|13.6|18.11|22.41|
> |Learnable node embedding (ours)|**0.01**|**0.04**|**0.1**|**0.19**|**0.31**|**0.44**|**0.6**|**0.73**|**0.93**|**1.14**|
>
> We admit that a C++ implementation is a valuable suggestion for the improvement of computational efficiency and will leave this for future work.

---

> > ### Author Response · Authors · 2024-11-23
> > **Official comment by authors (2/2)**
> >
> > **About taking $n'=d$ in message passing**
> > Thanks for the suggestion. First, we would like to clarify that due to the vectorized computation in pytorch, the dimension of embeddings is not a bottleneck for the computation efficiency of both ARTree and ARTreeFormer. In fact, in our implementation we take the feature dimension to $d=100$, which is larger than the embedding dimension $n$ (27~64) in all experiments. We also report the computation time for a smaller $d$ and $n'=d$ in the table above.
> > We see that a smaller $d$ may reduce the computation cost but is still much slower than ARTreeFormer.
> >
> > **About the backpropagation consumption**
> > In fact, there are three components that contribute to additional time consumption in practical training: gradient backpropagation, branch length generation, and likelihood evaluation. The likelihood evaluation part can depend on the specific data sets.
> > Therefore, we tested the practical training time of ARTree and ARTreeFormer in the benchmarks DS1-8, as shown in Figure 3 (right) and Figure 4 (right).
> >
> > To more directly reflect the time consumption of gradient computation, we report this in the following table. We see that ARTreeFormer still enables faster gradient computation.
> >
> > *Table: Time of gradient computation per 10 iterations on the TDE task.*
> > |Time (seconds) |DS1|DS2|DS3|DS4|
> > |---|---|---|---|---|
> > |ARTree|1.07|1.29|1.87|2.13|
> > |ARTreeFormer|**0.5**|**0.58**|**0.93**|**1.1**|
> >
> >
> > **Ablation study on the module combinations**
> > To inspect which component contributes to the reduced performance of ARTreeFormer, we conducted an ablation study. We see that the strong representation power of topological node embedding is the most crucial part of a model's performance. Importantly, the attention-based local message passing improves both accuracy and efficiency.
> >
> > *Table: KL divergence obtained by various model architectures on the TDE task.*
> > | Model | DS1 | DS2 | DS3 | DS4 |
> > | --- | --- | --- | --- | --- |
> > | Topological Node Embedding + Global GNN ( = ARTree) | 0.0045 | **0.0097** | 0.0548 | 0.0299 |
> > | Topological Node Embedding + Local Attention | 0.0089 | 0.0102 | **0.0441** | **0.0276** |
> > | Learnable Node Embedding + Global GNN | **0.0044** | 0.0123 | 0.0916 | 0.0346 |
> > | Learnable Node Embedding + Local Attention (= ARTreeFormer)| 0.0067 | 0.0102 | 0.0777 | 0.0320 |
> >
> > **About the complex conception in ARTreeFormer**
> > We have provided the Python implementation of ARTreeFormer which can be readily used for further applications.
> > Figure 1 has no cycles; the red arrows describe the message passing step, which is common in graph learning methods.
> > The model architecture of ARTreeFormer is not complex -- instead, we make a careful design that enables vectorized computation. Benefiting from this, the actual implementation of ARTreeFormer is highly organized and vectorized, which also greatly improves the computational speed.
> > See Figure 2 for a detailed runtime comparison of different modules.
> > In contrast, it is the inefficient design (although easy to describe) of ARTree, e.g., the multiple and irreducible loops (repetitive dirichlet energy minimization for topological node embeddings), that causes the highly inefficient computations. Therefore, a more efficient and faster design is very necessary.
> > See a more detailed discussion on page 6 (from line 301 to line 323).
> >
> > **About the misuse of words and vague description**
> > We apologize for this misuse of vectorization and parallelization. All the parallelization should be replaced by vectorization. We have revised our paper accordingly.
> > The qualitative claims of efficiency gain have been replaced by quantitative ones if applicable.
> >
> > **Response to your questions**
> >
> > (1) Exactly, we benchmark the Python implementation you cited above.
> >
> > (2) See the time complexity analysis above. We also provide a detailed analysis the revised manuscript.
> >
> > (3) The number of operations in topological node embedding is $O(N^3)$, but noting that the vectorized tensor operations in PyTorch are faster, the theoretical computational complexity in topological node embedding should lie between $O(N^2)$ and $O(N^3)$.
> > The empirical result aligns with this result.
> >
> > (4) We report the training time (which includes the gradient propagation) in Figure 3 (right) and Figure 4 (left). Please refer to "About the backpropagation consumption" for more details.
> >
> > (5) We apologize for the misuse of words. It should be "vectorization".
> >
> > (6) This is analyzed and visualized in [1] (see Figure 3 therein).
> >
> > (7) Thanks for your suggestions! We have implemented your suggested methods. Please check our response in "Topological embedding with fix-point algorithm" and "About taking $n'=d$ in message passing".
> >
> > [1] Chris Whidden and Frederick A Matsen IV. Quantifying MCMC exploration of phylogenetic tree space. Syst. Biol., 64(3):472–491, May 2015.

---

> > > ### Comment · Reviewer_3iPi · 2024-11-26
> > > **Reply to authors (1/2)**
> > >
> > > I thank the authors for the hard work they have put into incorporating my feedback into their manuscript. I think the work is heading in the right direction. However, I still have concerns:
> > >
> > > - The authors have confirmed in their response my suspicion that they have benchmarked a Python implementation of topological node embeddings in ARTree which is, from what I can tell, quite inefficient. As I said in my original review, it would not be surprising for a C++ implementation of topological node embeddings in ARTree to be >=10x faster. Since the speedup of ARTreeFormer over ARTree is 3-10x (depending on the task) and the paper repeatedly stresses that the topological node embedding module of ARTree is a key bottleneck, I find that the benchmarking against ARTree is rather unfair. ARTreeFormer's runtime (or, more specifically, its novel node embedding module's runtime) should be compared against an efficient implementation of topological node embeddings. Python is well-known to have massive overheads.
> > >
> > > - What model of computation are you using for describing computational complexity of the vectorized implementations? I have never seen this model with alpha representing the "accelerated order of vectorized linear operations" (Table 1, e.g. $O(N^{1 + \alpha} d ^{2 \alpha})$). Is this a standard model? Could you please describe or reference the model and provide examples of prior works using this model of computation?
> > >
> > > - Releated to the previous point: what I was expecting in terms of computational complexity was the total work required by the algorithm (e.g. $\Theta(N^3)$ for the topological node embeddings in ARTree) and a discussion of the degree of vectorization, such as "it is $\Theta(TN^3)$ where $T$ is the number of trees and $N$ the number of nodes, and cannot be vectorized on any dimension" or the opposite, such as "it is $\Theta(TN^3)$ and trees can be batched together along the first dimension, and furthermore the $N^3$ comes from matrix multiplies, which makes the algorithm specially well-suited for GPU".
> > >
> > > - I suggested that a vectorized implementation of ARTree's topological node embeddings module is possible with fixed-point iteration or repeated squaring, and the authors replied that (1) they tried power iteration and it is still slow, and (2) that "as the corresponding linear system is not homogeneous, the square trick cannot be applied". I really appreciate that the authors even tried this scheme. However: how did you implement fixed-point iteration? Can you please provide your implementation? If properly implemented, it can be vectorized over trees, and one can further exploit sparse matrix-vector multiplies. Without seeing the actual implementation it is hard to know whether it is inherently slow or was implemented inefficiently in Python. Second, the squaring trick can be applied:
> > >
> > > Consider the adjacency matrix $A$ of the graph $G$, which has size $(2n - 2) \times (2n - 2)$. Assume that the leaves are labeled $1$ to $n$ and the internal nodes $n+1$ to $2n-2$. Replace the first $n$ rows of $A$ by the first $n$ basis vector. Scale the remaining rows by $1/3$. Let this be $A'$. For example, for the tree with leaves $\{1,2,3,4\}$, internal nodes $\{5,6\}$ and edges $\{(1,5), (2,5), (5,6), (6,3), (6,4)\}$, we get:
> > >
> > > A' =
> > >
> > > 1     0     0     0     0     0
> > >
> > > 0     1     0     0     0     0
> > >
> > > 0     0     1     0     0     0
> > >
> > > 0     0     0     1     0     0
> > >
> > > 1/3  1/3  0     0     0  1/3
> > >
> > > 0     0      1/3  1/3  1/3  0
> > >
> > > Given the leaf embeddings $V_L \in R^{n \times d}$ note that $V_I \in R^{(n - 2) \times d}$ is a solution to the Dirichlet energy minimization problem if and only if $A' [V_L; V_I] = [V_L; V_I]$ (i.e. each internal node's embeddings is the average of the neighbors' embeddings). Fixed-point iteration becomes the (sparse) matrix-vector multiply:
> > >
> > > $[V_L; V_I^{(t + 1)}] = A' [V_L; V_I^{(t)}]$
> > >
> > > which can be batched across trees. Runtime to get $V_I^{(g)}$ is $O(g n d)$ where $g$ is the number of iterations. It is worse than the $O(n d)$ two-pass algorithm, but vectorizable in several dimensions. Repeated squaring means:
> > >
> > > $[V_L; V_I^{(2^p)}] = (A'^{2^p}) [V_L; V_I^{(0)}]$
> > >
> > > which can also be batched across trees and involves repeated matrix-matrix multiplies. Runtime to get $V_I^{(g)}$ for $T$ trees is $O(T n^3 \log g + T n^2 d)$ which looks terrible but these are all highly vectorized operations which GPU particularly excels at. In fact, if I understood your model of computation, the runtime of repeated squaring would be something like $O(T^\alpha n^{3\alpha} \log g + T^\alpha  n^{2\alpha} d^{\alpha})$ which looks very fast.
> > >
> > > [Continued in the next comment]

---

> > > > ### Comment · Reviewer_3iPi · 2024-11-26
> > > > **Reply to authors (2/2)**
> > > >
> > > > - I would like to point out that the two implementations above enable learning the leaf embeddings $V_L$ (instead of fixing them to one-hots) since unlike ARTree's original implementation, one can easily differentiate wrt $V_L$ (it's just a matrix multiply layer to get $V_I$ from $V_L$). I think this is an interesting direction for future work which increases interest in topological node embeddings.
> > > >
> > > > - If desired one can analyze the convergence rate of the fixed-point iteration scheme by noting that fixed-point iteration is equivalent to gradient descent on the Dirichlet energy function with a learning rate of $1/3$. Without going into too much detail (I just sketched it out), and at the risk of making mistakes: the Dirichlet energy function scaled by $1/2$ can be written as $f(x) = (1/2) x^T S x + b^T x + c$ where $S$ is the graph Laplacian of $G$ restricted to the internal nodes and $b, c$ are constants. Gradient descent on a function of the form $f(x) = (1/2)x^T S x + b^T x + c$ with positive definite $S$ and optima $x^*$ is well-understood: the updates satisfy $(x^{t+1} - x^*) = (I - \alpha S)(x^t - x^*)$, so that the squared error contracts by at least $|I - \alpha S |_\text{op}$ each step. A typical tree also has diameter $O(\log n)$, not $O(n)$, so $O(\log n)$ iterations of fixed-point iteration may suffice. With repeated squaring the are really no concerns about convergence speed.
> > > >
> > > > To summarize: I find the claim that topological node embeddings are slow is not accurate. A more efficient implementation outside of pure Python has not been benchmarked. Since the slowness of topological node embeddings is one of the main points the work hinges on, I will maintain my original score.

---

> ### Author Response · Authors · 2024-11-27
> **Response to the follow-up questions (1/2)**
>
> Thanks for your follow-up questions. Before addressing your concerns, we think we should first establish a consensus that **two models should be compared in the same programming language, the same device, and possibly, the same embedding dimension.**
>
> - Firstly, we want to clarify your potential misunderstanding of our acceleration claim. The '3$\times$' is the acceleration of training (training includes running the whole deep model, backpropagation, and likelihood computation), and '10$\times$' is the acceleration of generating (including inference of the whole deep model). In both cases, **topological node embedding is only a part of the whole process.** If you are talking about the whole process and the acceleration of the C++ implemented ARTree is 10x, then 10x acceleration should be the same for C++ implemented ARTreeFormer.
> Now we only talk about the implementation of the topological node embedding module.
>
> (a) We assume the topological node embedding of ARTree is computed by the two-pass algorithm, and defer the discussion on other choices in the following points. In the Python implementation, the acceleration of ARTreeFormer upon the current two-pass algorithm of ARTree is $20\times$. A potential overhead in Python is the tree traversal implementation, which we reported below. Compared to the full node embedding module, we see that **the pure traversal of a tree is negligible in ARTree in the Python implementation**, so it is the numerical computation that dominates the runtime, not the tree traversal.
>
> *Table: Computation time (seconds)*
> |Method\Number of leaves |10|20|30|40|50|60|70|80|90|100|
> |---|---|---|---|---|---|---|---|---|---|---|
> |Topological node embedding with two-pass algorithm|0.18|0.78| 1.97|3.33|5.31|7.52|10.58|13.6|18.11|22.41|
> |**Only two-pass traversal without any numerical computation**|0.02|0.07|0.15 |0.27|0.42|0.6|0.82|1.07|1.36|1.69|
> |Learnable node embedding (ours)|**0.01**|**0.04**|**0.1**|**0.19**|**0.31**|**0.44**|**0.6**|**0.73**|**0.93**|**1.14**|
>
> (b) In a C++ implementation, the two for-loops (iteration over all immediate subtrees; iteration over all nodes) still cannot be reduced. We apologize for not being able to deliver a C++ runtime comparison given the limited time budget. However, as PyTorch is based on C++ backend, we expect the ratio of runtimes of ARTreeFormer and ARTree implemented in C++ to be similar to what we observed on the Python side, that is (runtime of ARTreeFormer in C++)/(runtime of ARTree in C++) $\approx$ (runtime of ARTreeFormer in Python)/(runtime of ARTree in Python).
>
> *Table R1: Time consumption (seconds) of node embedding module in generating 100 trees. ARTree's default setting means the embedding dimension is $N$.*
> |Method\Number of leaves |10|20|30|40|50|60|70|80|90|100|
> |---|---|---|---|---|---|---|---|---|---|---|
> |Topological node embedding with fix-point algorithm without square trick (ARTree's default setting, tol=1e-5)|0.07|0.28|0.61 |1.12|1.92|3.07|4.46|6.38|8.78 |11.72|
> |Topological node embedding with fix-point algorithm with square trick (ARTree's default setting, tol=1e-5)|0.05|0.16|0.35|0.62|1.03|1.63|2.45|3.6|5.15|7.13|
> |Topological node embedding with two-pass algorithm|0.18|0.78| 1.97|3.33|5.31|7.52|10.58|13.6|18.11|22.41|
> |Learnable node embedding (ours)|**0.01**|**0.04**|**0.1**|**0.19**|**0.31**|**0.44**|**0.6**|**0.73**|**0.93**|**1.14**|
>
> - We use the $\alpha$ notation here to reflect the acceleration achievable by vectorized computation. As different programming languages or computation module may lead to different vectorization efficiency, we use a flexible $\alpha$ here. If one wants to see how the model can be "perfectly vectorized", just let $\alpha=0$.
>
> - Thanks for this example. We would like to give the following explanations for the complexity of ARTree and ARTreeFormer. For generating one tree with $N$ leaves:
>     - The time complexity of the node embedding module of ARTree is $O(N^3)$, where the third dimension can be vectorized; The time complexity of the node embedding module in ARTreeFormer is $O(N^2d+Nd^2)$ where for both terms, the second and third dimensions can be vectorized.
>     - The time complexity of the message passing module of ARTree is $O(N^2d^2)$, where the 2-nd, 3-rd, and 4-th dimensions can be vectorized. The time complexity of the message passing module of ARTreeFormer module is $O(Nd^2)$, where the 2-nd and 3-rd dimensions can be vectorized.
>
>     We see that the main difference lies in the node embedding module: ARTree has two dimensions that cannot be vectorized; ARTreeFormer only has one dimension that cannot be vectorized. When vectorization is enabled, the complexity of node embedding module would be dominant for both ARTreeFormer and ARTree.

---

> > ### Author Response · Authors · 2024-11-27
> > **Response to the follow-up questions (2/2)**
> >
> > - First, here is our code for implementing the fix-point iteration. In each iteration, we gather the node features from the neighborhood of a node and assign the average to this node. This function enables vectorization across different trees in a batch and different nodes on a tree. We want to emphasize that the fixpoint() should be called $(N-3)$ times, as there are $(N-3)$ immediate subtrees during the sequential generating process of the autoregressive model.
> > ```
> > def fixpoint(edge_index, X, tol=1e-5, max_iters=10000):
> >     # edge_index describes the index of the adjacency nodes of a node. Its shape is (number of trees, number of nodes, 3).
> >     # X is the initialization of the features of the nodes.
> >     # (each row of X is initialized as [1/feature_dim]*feature_dim)
> >
> >     bs, dim, nf = X.shape
> >     identity = torch.eye(dim+2, nf).unsqueeze(0).repeat(bs, 1,1)
> >
> >     for it in range(1, max_iters + 1):
> >         X_old = X.clone()
> >         neigh = torch.gather(torch.concat([identity, X], dim=1), dim=1, index=edge_index.reshape(bs, -1).unsqueeze(-1).repeat(1,1,nf))
> >         neigh = neigh.reshape(bs, dim, 3, nf)
> >         X = torch.mean(neigh, dim=2)
> >         Lnorm = torch.mean(torch.abs(X-X_old), dim=(1,2))
> >
> >         if not torch.any(Lnorm > tol):
> >             break
> >     return X
> > ```
> > Second, we really appreciate your instruction on implementing the square trick. Following your instructions, we implemented it and reported the computation time in Table R1. We see it indeed improves the computation efficiency upon the fix-point method without the square trick, but still fall quite behind the learnable node embedding module in ARTreeFormer. Moreover, regarding the complexity, we want to clarify that:
> >
> > (a) Although $A'$ is a sparse matrix, **$(A')^n$ becomes dense when $n$ gets large**. Therefore, we cannot expect to apply the sparse matrix multiplication when computing $(A')^n\times (A')^n$. That is to say, the square trick would be able to reduce the number of iterations, at the price of increasing the computation cost at each iteration, especially for large $n$.
> >
> > (b) In our original implementation of the fixpoint algorithm, we only have to average the neighboring node features, with a complexity of $O(N^2)$ in each iteration; For implementing the square trick, matrix multiplication with a complexity of $O(N^3)$ is required in each iteration. Finally, we sincerely appreciate the reduction of number of iterations from $O(g)$ to $O(\log g)$.
> >
> > Finally, we want to point out a potential missing point in your computation. The fixpoint algorithm should be called for $N-3$ times, i.e., iteration over $(N-3)$ immediate subtrees in the autoregressive growth. So the time complexity using fixpoint (with square trick) should be $O(NT^\\alpha N^{3\\alpha}\\log g)$, where the first $N$ cannot be vectorized.
> >
> > - Thank you very much for suggesting the usage of $V_L$ for incorporating the learnable leaf node features. We agree this is an interesting direction for future work.
> >
> > - In our opinion, $O(log n)$ is the depth of a balanced bifurcating tree of $n$ leaf nodes. However, there is no guarantee that phylogenetic trees will be balanced. In contrast, it is generally skewed so the depth should lie between $O(log n)$ and $O(n)$. From another point of view, the tree shape is related to the eigenvalues of the $S$ matrix and further affects the convergence rate of the fix-point algorithm.
> >
> > To summarize: we identify that topological node embeddings in ARTree require at least two for loops (one for iterating over the subtrees, another for iteration over the nodes for each of these subtrees) that are not vectorizable, which leads to the $O(N^2)$ complexity that are true for all programming languages. That complexity may be reduced to $O(N\log(N))$ with the square trick. However, the involved computation would become more expensive as the sparsity would be reduced as the power of $A'$ increases.
> > As pytorch has a C++ backend, and the tree traversal overhead is negligible, we believe that the ratio of runtimes of ARTreeFormer and ARTree would be similar, regardless of the programming languages. See Table R1 for detailed evaluation results.
> >
> > We hope our response adequately addresses your concerns and look forward to any further questions.

---

> > > ### Comment · Reviewer_3iPi · 2024-12-02
> > > **Response to the follow-up responses (1/2)**
> > >
> > > I thank the authors for following up on my response and providing me with further datapoints to base my review on.
> > >
> > > Regarding the comment "two models should be compared in the same programming language": This comment overlooks the nuances underlying different programming languages. While pure Python is very slow, many Python libraries like PyTorch and numpy are implemented in efficient programming languages like C++ under the hood. Now, if one takes a look at ARTree's modules, one notes that:
> > > (1) The topological node embedding module uses pure Python for tree traversal (as I have cited above in my original review).
> > > (2) The GNN is implemented in PyTorch.
> > > Thus, while both modules are implemented in Python, the overhead (i.e. constants) associated with each implementation are not comparable. Recall that the computational complexity of topological node embeddings step (1) is O(N^3), and of message passing step (2) is O(N^2d^2). With $N=1000$ and $d=100$ we see that topological node embeddings are doing ~10x less work than message passing, so how is it a bottleneck in practice? Let's do a back-of-the-envelope calculation based on real provided runtimes: in the revised version of the manuscript the authors remark that "ARTree [...] more than 10 seconds for generating 60 100-leaf trees". With a time complexity of O(#trees * N^3), the topological node embedding module's two-pass algorithm is performing ~ 60 * 100^3 = 6 x 10^7 operations. A back of the envelope calculation using 10^8-10^9 operations per second for an efficient C++ implementation gives around 0.1-1 seconds in this example. In practice, it took around 5 seconds (65% of 10s) i.e. around 10x more. This is consistent with the overhead of pure Python.
> > >
> > > The authors then state that: "If you are talking about the whole process and the acceleration of the C++ implemented ARTree is 10x, then 10x acceleration should be the same for C++ implemented ARTreeFormer." This is again inaccurate because in fact, ARTreeFormer uses PyTorch for all its modules, which is *already* implemented in an efficient programming language such as C++ under the hood. Thus, is it not possible to speed up ARTreeFormer this way. In constrast, ARTree's node embedding module appears to have significant room for speedup by using e.g. C++ instead of pure Python.
> > >
> > > Regarding: "Compared to the full node embedding module, we see that the pure traversal of a tree is negligible in ARTree in the Python implementation, so it is the numerical computation that dominates the runtime, not the tree traversal.". Without providing the profiling code I cannot determine whether the way it was profiled supports this claim. The Python implementation in ARTree contains not only tree traversal and vector operations, but also indexing of other structures such as a dictionary with string keys in `name_dict[node.name] = node`. Indexing a Python dictionary with a string key at each node visit may be problematic. Is this considered tree traversal runtime or numerical computation when you profiled the code?
> > >
> > > I am quite confused when the authors make the following statement: "as PyTorch is based on C++ backend, we expect the ratio of runtimes of ARTreeFormer and ARTree implemented in C++ to be similar to what we observed on the Python side, that is (runtime of ARTreeFormer in C++)/(runtime of ARTree in C++) ~ (runtime of ARTreeFormer in Python)/(runtime of ARTree in Python).". Following your notation, in your work you have benchmarked:
> > > - (runtime of ARTree's node embedding module in Python)
> > > - (runtime of ARTree's message passing module in PyTorch (C++))
> > > - (runtime of ARTreeFormer's node embedding module PyTorch (C++))
> > > - (runtime of ARTreeFormer's message passing module in PyTorch (C++))
> > >
> > > which again, in my opinion makes the benchmark unfair against topological node embeddings.
> > >
> > > Regarding the use of $\alpha$ notation in the revised work to reflect the acceleration achievable by vectorized computation: the authors have failed to provide a reference to this model of computation or to prior works using it, which suggests to me that the authors have invented it. This in turn suggests to me that the authors may be unfamiliar with models of computation and high performance computing more broadly, which are quite important topics for the kind of research that is being conducted here.
> > >
> > > Regarding "we want to point out a potential missing point in your computation": I apologize if my writing caused confusion, I was referring to the computational complexity of a single step of topological node embeddings with batched trees. This is why I used lowercase $n$ instead of uppercase $N$. I agree one needs to multiply by N to get the total runtime for generating a full tree. Indeed, all methods have this leading $N$ term from the sequential leaf adding. This potential confusion does not change my point.
> > >
> > > [Continued.]

---

> > > > ### Comment · Reviewer_3iPi · 2024-12-02
> > > > **Response to follow-up responses (2/2)**
> > > >
> > > > Regarding the fixed point iteration method: While it is true that fixed point iteration would in general require $\Omega(n)$ iterations to "propagate" information from one side of the tree to another when the diameter of the tree is $\Omega(n)$, topological node embeddings decay exponentially fast with distance. Indeed, as lemma 1 from [Zhang 2023, "Learnable Topological Features for Phylogenetic Inference via Graph Neural Networks"] implies, if the distance between source leaf $l$ and internal node $i$ is $d$, then ARTreeFormer's topological node embeddings satisfy $v_{li} \le 2^{-d}$. Intuitively, topological node embeddings look like exponentially decaying bump functions centered around each leaf, much like partitions of unity. Thus, even for graphs with diameter $\Omega(n)$, fixed point iteration with $O(\log n)$ iterations may converge very fast, since far away nodes have embedding coordinates of near-zero anyway. For general embeddings (not just one-hot), the same intuition should hold: each leaf only really influences the embeddings of nodes within a $O(\log n)$ radius.
> > > >
> > > > I ran your provided fixed-point iteration code on a "chain tree" with as many as 1000 leaves and it converges in around 20 iterations (tol=), which seems to validate my observation in the previous paragraph. The code and runtimes you provided in table R1 for fixed-point iteration look reasonable to me: a ballpark calculation for N=100, num trees=100, and 20 iterates gives me N^3 * num_trees * iterations = 100^3 * 100 * 20 = 2 * 10^9 operations, i.e. around 2-20 seconds for an efficient C++ implementation using 10^8-10^9 operations per second. Your table shows 11.72s, which falls exactly within this range. While this implementation does not shine on table R1 (ARTreeFormer's learnt node embeddings are around 10x faster), it is important to remark that it *is* a vectorized implementation and thus might excel on e.g. GPU. So, vectorized implementations of topological node embeddings are possible and reasonable. But onto the most problematic point: table R1 shows that topological node embeddings with the linear-time two-pass algorithm takes 22.41s. This does not match the back of the envelope calculation: the computational complexity gives around num_trees * N^3 = 100 * 100^3 = 10^8 operations, which gives between 0.1-1s for an efficient C++ implementation, i.e. >=10x faster. This is reasonably explained, again, by the use of an inefficient Python implementation for topological node embeddings.
> > > >
> > > > **To summarize**: The performance benchmark of ARTree's topological node embedding module against ARTreeFormer's remains unfair in my opinion, since ARTree's module is implemented in pure Python while all other modules in ARTreeFormer are implemented in PyTorch (and thus under the hood in C++). In fact, by taking a look at the runtimes reported by the authors and comparing them to back-of-the-envelope calculations I found further evidence that the overhead incurred by the topological node embedding's Python implementation may be quite high (>=10x). In addition, the author's use of what appears to be invented, non-standard notation to describe the speedups of vectorized implementations suggests to me that the authors may be unfamiliar with models of computation and high performance computing more broadly, which are quite important topics for the kind of research that is being conducted here. Overall, I continue to recommend rejection. I want to stress that the manuscript contains many valuable ideas, but in its current form the discussion on runtime of topological node embeddings is quite problematic.

---

> ### Author Response · Authors · 2024-12-04
> **Further clarification**
>
> We thank you for your follow-up questions. Let us focus on several notable points for a more precise understanding of the implementation and time consumption of ARTree/ARTreeFormer.
>
> Your comment "ARTree's node embedding module in Python" is inaccurate. All modules of ARTree, including the node embedding module, are implemented in PyTorch, not pure Python.
> One can tell this by simply looking at the ```torch.Tensor``` type of the ```leaf_features``` in the following code block.
> We have also profiled the node embedding module of ARTree, and the results have been shown below.
> We see that a large amount of time (around $55\\%$) is spent on the PyTorch-based numerical computation, which should have a similar consumption in C++.
> This is to say, although the overhead of the pure Python part makes the current implementation of ARTree slow, the main part of the node embedding module in ARTree (which is written using Pytorch that has C++ under the hood) can not be sped up by using C++.
> Therefore, **ARTree's node embedding module does not have much room for speedup by using C++ (at most 2 times faster based on our profiling).**
>
> Regarding your example, we would like to point out that **all the runtime is reported with vectorized computation enabled in pytorch**. Therefore, your simple calculation of time complexity is misleading. With vectorization, the time complexity of topological node embedding step (1) is roughly $O(N^2)$, and message passing step (2) is $O(N)$.
> Therefore, the node embedding step of ARTree can be a bottleneck in practice as demonstrated in our experiments. In contrast, with vectorization, the node embedding module of ARTreeFormer can be much faster. This is the main reason behind the results in Table R1.
> For your back envelop time calculation, we'd like to remind you that the absolute speed depends on the computing hardware and the big $O$ notation omits the constant term. Therefore, only the relative speed can be fairly compared.
>
> Your table of benchmarked runtime is also misleading. As mentioned above, the node embedding module of ARTree also uses Pytorch. The overhead of Python is at most half the computation. The numerical computation part (coded in Pytorch) takes at least half of the runtime reported in Table R1, and the acceleration of the learnable node embedding module in ARTreeFormer remains significant.
>
> - (runtime of ARTree's node embedding module in Pytorch (C++))
> - (runtime of ARTree's message passing module in PyTorch (C++))
> - (runtime of ARTreeFormer's node embedding module PyTorch (C++))
> - (runtime of ARTreeFormer's message passing module in PyTorch (C++))
>
> Regarding the fixed point iteration method: We'd like to point out that **the topological embeddings do not decay exponentially fast with distance**, as there is also a constant term $d_u$ there for the update equation. In fact, all the topological node embeddings after convergence lie in the simplex, that is **the topological node embedding $x_u$ of node $u$ satisfies $\sum_{n=1}^N x_u[n]=1$, where $x_u[n]$ is the $n$-th element of $x_u$**, as suggested by Theorem 1 in [Zhang, 2023]. Therefore, for graphs with diameter $\Omega(n)$, fixed point iteration with $O(\log n)$ iterations can still be far away from convergence. And those nodes with embedding coordinates of near-zero would be incapable of representing local geometric structures of the tree topology, leading to a potential drop in performance.
>
> Regarding your example:
> With perfect vectorization, the time complexity of fix-point algorithm should be $O(N * \log N)$ (the best case of a balanced tree), while that of the two-pass algorithm should be $O(N^2)$.
> Now consider one tree with $N=100$ leaf nodes, and 20 iterations are required for the convergence of the fixed point algorithm.
> With vectorization, the time complexity of fix-point algorithm should be $C_1\cdot 100\cdot 20$, while that of the two-pass algorithm should be $C_2 \cdot 100^2$.
> This complexity ratio aligns with our empirical results.
>
>
> **To summarize**: We have done an additional profiling of ARTree's node embedding module. We found that the overhead of Python takes at most half the runtime. This means that the numerical computation in Pytorch takes at least one half of the runtime, which can not be reduced much when using C++. Therefore, the claim that the overhead of Python is >=10x is inaccurate. Moreover, for the runtime analysis, we want to point out that those runtimes are all obtained with vectorized computation enabled through Pytorch. Therefore, a more appropriate analysis should use the corresponding vectorized complexities instead of the standard ones (which were used by the reviewer). As shown in the discussion above, our runtime results align with the vectorized complexities of ARTree and ARTreeFormer. We sincerely hope these clarifications can address your concerns and help understand our time consumption better.

---

> > ### Author Response · Authors · 2024-12-04
> > **Profile of ARTree's node embedding module**
> >
> > ```
> > Line #      Hits         Time  Per Hit   % Time  Line Contents
> > ==============================================================
> >     35                                           @profile
> >     36                                           def node_embedding(tree, level):
> >     37     10000       5253.0      0.5      0.0      name_dict = {}
> >     38     10000       4410.0      0.4      0.0      j = level
> >     39     10000      46496.0      4.6      0.1      rel_pos = np.arange(max(4,2*level-4))
> >     40   1990000    6115957.0      3.1      9.5      for node in tree.traverse('postorder'):
> >     41   1980000    1536086.0      0.8      2.4          if node.is_leaf():
> >     42   1000000     291708.0      0.3      0.5              node.c = 0
> >     43   1000000    2545394.0      2.5      3.9              node.d = leaf_features[node.name]
> >     44                                                   else:
> >     45    980000     947045.0      1.0      1.5              child_c, child_d = 0., 0.
> >     46   2950000    1344661.0      0.5      2.1              for child in node.children:
> >     47   1970000     670931.0      0.3      1.0                  child_c += child.c
> >     48   1970000    7407605.0      3.8     11.5                  child_d += child.d
> >     49    980000     499189.0      0.5      0.8              node.c = 1./(3. - child_c)
> >     50    980000    5303259.0      5.4      8.2              node.d = node.c * child_d
> >     51    980000     697394.0      0.7      1.1              if int(node.name) < len(rel_pos):
> >     52    960000     563908.0      0.6      0.9                  rel_pos[node.name] = j
> >     53    980000     485237.0      0.5      0.8              node.name, j = j, j+1
> >     54   1980000     988536.0      0.5      1.5          name_dict[node.name] = node
> >     55
> >     56     10000       4767.0      0.5      0.0      node_features, node_idx_list, edge_index = [], [], []
> >     57   1990000    4912675.0      2.5      7.6      for node in tree.traverse('preorder'):
> >     58   1980000     443753.0      0.2      0.7          neigh_idx_list = []
> >     59   1980000    1597575.0      0.8      2.5          if not node.is_root():
> >     60   1970000   17219507.0      8.7     26.6              node.d = node.c * node.up.d + node.d
> >     61   1970000    1596294.0      0.8      2.5              neigh_idx_list.append(node.up.name)
> >     62
> >     63   1970000    1758935.0      0.9      2.7              if not node.is_leaf():
> >     64    970000    2501343.0      2.6      3.9                  neigh_idx_list.extend([child.name for child in node.children])
> >     65                                                       else:
> >     66   1000000     542745.0      0.5      0.8                  neigh_idx_list.extend([-1, -1])
> >     67                                                   else:
> >     68     10000      23428.0      2.3      0.0              neigh_idx_list.extend([child.name for child in node.children])
> >     69
> >     70   1980000     894422.0      0.5      1.4          edge_index.append(neigh_idx_list)
> >     71   1980000     762492.0      0.4      1.2          node_features.append(node.d)
> >     72   1980000     770827.0      0.4      1.2          node_idx_list.append(node.name)
> >     73
> >     74     10000     522331.0     52.2      0.8      branch_idx_map = torch.sort(torch.tensor(node_idx_list).long(), dim=0, descending=False)[1]
> >     75     10000    1082825.0    108.3      1.7      edge_index = torch.tensor(edge_index).long()
> >     76
> >     77     10000     539947.0     54.0      0.8      return torch.index_select(torch.stack(node_features), 0, branch_idx_map), edge_index[branch_idx_map], torch.from_numpy(rel_pos), name_dict
> > ```

---

### Official Review · Reviewer_ogwQ · 2024-11-04

**Soundness:** 4
**Presentation:** 3
**Contribution:** 2
**Rating:** 8
**Confidence:** 2

**Summary:**

The authors present **ARTreeFormer**, an improvement on a previous method: **ARTree**.
 These methods are autoregressive GNN-based models and allow for a flexible way to model the entire tree space. Both **ARTree** and **ARTreeFormer** can be used in Bayesian phylogenetic tasks, including variational Bayesian phylogenetic inference.

The core methodological aspects of both methods are the same: starting with the simplest tree topology $\tau_3$ with 3 nodes, they iteratively add leaf-nodes to the tree by selecting edges on which to attach them until all $N$ leaf nodes are contained in the tree.
At a step $n<N$, a probability vector $q_n\in\mathbb{R}^{|E_n|}$ over the set of edges of the tree is computed using a GNN, and the edge on which to attach the $(n+1)^{th}$ leaf node is sampled from $E_n$ according to $q_n$.
The probability vector $q_n$ is computed from edge embeddings which, in turn, are computed from tree *node embeddings* after several rounds of *message passing*.

The latter two steps are where the two methods differ and allow **ARTreeFormer** to be faster than **ARTree**:
 1. *Node embedding*: **ARTreeFormer** uses the very popular self-attention mechanism to learn topological embeddings for tree nodes. This eliminates a tree-traversal based algorithm used in **ARTree** to compute these embeddings allowing for **(1)** faster execution as self-attention can be efficiently implemented in a vectorized fashion and **(2)** batched parallel execution of this embedding step on several topologies at once.
 2. *Message passing*:  **ARTreeFormer** uses *local* message-passing to update node embeddings after addign a new leaf-node, where only the nodes in the direct neighbourhood of the newly-created internal node are updated. This contrasts with the approach in **ARTree** where all the nodes in the graph would be updated at each step of the iterative algorithm.

These two modifications allow **ARTreeFormer** to consistently run faster than **ARTree** without sacrificing performance.

**Strengths:**

1. Overall the paper is fairly well-written and flows quite well
2. The learnable node embeddings using attention are a simple but effective way to **(1)** make the embeddings and the model more flexible **(2)** improve the runtime
3. There is a significant improvement in runtime shaving *hours* of CPU runtime off (e.g for the Maximum parsimony application). I believe this makes the method more applicable to modern phylogenetic datasets where there are several hundred/thousands of taxa.

**Weaknesses:**

The paper, I believe, has no real "deal-breaking" weakness but would benefit from addressing the following points:

1. In the VBPI experiment, **ARTreeFormer** is compared to other methods: $\phi$-CSMC and GeoPhy in terms of approximation accuracy. However, in terms of runtime, **ARTreeFormer** results are only compared to methods upon which it was built *(i.e. SBNs and ARTree)*. Since the focus of this paper is the fast runtime of **ARTreeFormer** I think a computation-speed comparison to all methods is warranted.
2. While I understand the motivation behind choosing a one dimensional query vector $q_n$ in the node embedding phase *(a lowered time complexity and runtime)* , does that not limit the expressivity of the model ? It is not evident to me that each node benefit from querying the same node features.
3. If the main benefit of **ARTreeFormer**  is a decreased runtime, why not explore alternatives to VIMCO to estimate gradients of the tree topology parameters ? As I understand it this approach requires $K$ samplings of the topology distribution to estimate gradients. It might be faster to compute gradients using methods like CONCRETE [1] or the Gumbel-softmax trick [2] to backpropagate directly through the discrete iterative tree building algorithm.

[1] https://doi.org/10.48550/arXiv.1611.00712
[2] https://doi.org/10.48550/arXiv.1611.01144


*Minor:*
-  In (12), should it be $\ell(f_n,\tau_{n+1},w):=\sum_{(u,w)\in E_{n+1}} ||f_n(u) - f_n(w)||^2$ ? i.e. $f_n(w)$ instead of $f_n(v)$

**Questions:**

1. Could the authors try and compare the runtime of **ARTreeFormer** to independent methods as mentioned in weakness 1 ?
2. The paper might benefit form another ablation study, (similar to what has already been done for $K$ and $d$ in appendix $E$), to quantify the impact of the choice of a 1D query vector as mentioned in weakness 2.

---

> ### Author Response · Authors · 2024-11-23
> **Official comment by authors**
>
> Thanks very much for your constructive review and positive rating of our paper. Here are our responses:
>
> ## Weaknesses
>
> **(1)**: In the VBPI experiment, ARTreeFormer is compared to other methods (...)
>
> **Response**:
> We have supplemented the training time of GeoPhy in Table 8, and also put it here for your convenience.
> $\phi$-CSMC is an SMC-based method that does not require training but may cost a large amount of time for inference (more than 10 minutes for sampling 100 trees), and therefore is not considered here.
> We'd like to emphasize that although ARTreeFormer can cost more time than GeoPhy, it achieves much better approximation accuracy.
>
> *Table: Training time (seconds) per passing 100 trees*
> |Method| ARTreeFormer | ARTree | SBN | GeoPhy |
> | ---|---|---|---|---|
> |DS1 (27 leaf nodes) | 2.49| 6.06 | 1.02 | 1.87 |
> |DS8 (64 leaf nodes) | 7.27|26.77 | 2.05 | 2.90 |
>
>
> **(2)**: While I understand the motivation behind choosing a one dimensional query vector in the node embedding phase (...)
>
> **Response**: Thanks for your question! Note that we want to compute a one-dimensional feature vector for the newly added nodes, therefore one dimensional query is enough in terms of model design.
> This choice also follows [Han et al, 2023] who also considers one-dimensional query vector for forming node features.
> Employing a multi-dimensional query certainly also works for this problem, and we reported the results in the following Table, where we see that the effect of the additional query vectors is negligible.
>
> |Number of queries \ Data set|DS1|DS2|DS3|DS4|
> |---|---|---|---|---|
> |1|0.0067|0.0102|0.0777|0.0320|
> |4|0.0064|0.0102|0.0763|0.0318|
> |16|0.0065|0.0101|0.0772|0.0325|
>
> **(3)**: If the main benefit of ARTreeFormer is a decreased runtime, why not explore alternatives to VIMCO (...)
>
> **Response**:
> Thanks for your suggestion!
> As can be seen in Table 8 in our revision (or in our response to (1)), VIMCO is not the bottleneck for the computation efficiency of ARTree, as it is also used in SBN which is much faster than the other methods.
> The inefficiency of ARTree mainly comes from the repetitive non-vectorizable topological node embedding and the global message passing scheme, as discussed in Section 3.1.
> Alternative gradient estimation methods like CONCRETE [1] or Gumbel-softmax trick [2] have also been used for discrete models. However, the main difficulty of applying these gradient estimators to phylogenetic models is that the phylogenetic likelihood function depends on the tree topology in a rather subtle way, which makes it hard to differentiate through the discrete iterative tree building algorithm.
> Thanks again for this interesting suggestion and we will explore this in future work.
>
>
> **Minor**: In (12), should it be (...)
>
> **Response**: Thanks! We have fixed it in our revision.
>
>
> ## Questions
>
> **(1)**: Could the authors try and compare the runtime of ARTreeFormer to independent methods as mentioned in weakness 1?
>
> **Response**: Thanks for this question. Please refer to our response to weakness 1.
>
> **(2)**: The paper might benefit form another ablation study, (similar to what has already been done for $K$ and $d$ in appendix), to quantify the impact of the choice of a 1D query vector as mentioned in weakness 2.
>
> **Response**: Please refer to our response to weakness 2.

---

> > ### Comment · Reviewer_ogwQ · 2024-11-26
> >
> > I thank the authors for their response.
> >
> > Overall, I believe that they have sufficiently addressed the concerns I raised in my initial review.
> > The modifications made in response to comments from other reviewers also improve the readability and clarity of the paper.
> >
> > I do not think it necessary to change my rating.

---

### Official Review · Reviewer_5tHT · 2024-11-09

**Soundness:** 4
**Presentation:** 3
**Contribution:** 3
**Rating:** 8
**Confidence:** 2

**Summary:**

The paper introduces ARTreeFormer, which aims to improve computational efficiency in phylogenetic inference by leveraging attention mechanisms. ARTreeFormer is based on ARTree. For traditional ARTree, the total run time grows rapidly and the
node embedding module dominates the total time (the message passing layers are relatively short in time consumption). In comparison to ARTree, ARTreeFormer uses (1) the recurrent node embedding that is learnable through a deep graph neural network layers (2) a local updating scheme in the neighborhood of the newly added internal node, instead of a global one. In the experiment, ARTreeFormer is evaluated with tree topology density estimation (TDE) and variational Bayesian phylogenetic inference (VBPI) on DS1-8. Experiments also show that ARTreeFormer is significantly faster than ARTree in training and evaluation.

**Strengths:**

(1) ARTreeFormer improves significantly over ARTree by recognizing the several design bottlenecks and running time complexity of ARTree. The ARTreeFormer proposes novel attention and GNN mechanism that is tailored to phylogenetic inference.

(2) ARTreeFormer is evaluated on several standard phylogenetic inference benchmarks, while the authors have provided several useful metrics in analyzing the performance of the model. The author explains in details about what each metrics stand for, and make insightful comments on why ARTreeFormer is suprior to SOTA methods in certain metrics.

(3) The paper is written with a logical flow and clear exposition, and the authors provide an algorithm pseudo codes that makes the algorithm easy to understand.

(4) The ARTreeFormer's performance in FLOPs and time complexity seem to significantly improve over ARTree, which showcase the effectiveness of the proposed algorithm.

**Weaknesses:**

(1) Table 1 reveals that ARTreeFormer consistently lags behind ARTree in terms of KL divergence. Why does ARTreeFormer not achieve competitive performance with ARTree, which it was designed to improve? What training components or procedures might contribute to ARTreeFormer’s reduced performance?

(2) It seems that the recurrent node embedding with simplified attention mechanism and local message passing updates module are built upon two separate components of ARTree, respectively. The authors should consider to bring an ablation study on only changing one of the modules for ARTree and observe the metrics (time,FLOP, KL divergence, etc.). Since attention mechanisms are a key part of ARTreeFormer’s design, the paper could also have a deeper ablation of different attentions.

(3) There is not a theoretical analysis on how ARTreeFormer improves over ARTree in terms of running time. While the authors mention that ARTree builds topological node embeddings with two-pass algorithm which needs additional traversal over tree topology, the authors could mention how big-O complexity of the ARTree is for node-embedding and message passing module. Then, it would also be beneficial to include the recurrent node embedding and local message passing updates' complexity in terms of big-O and model sizes. Overall, the authors could make a more compelling argument than showing Figure 2 (and give a roughly estimate like 65%).

(4)  Small thing on the writing, but the authors spend two paragraphs explaining the VBPI score and parsimony score before introducing the ARTree model, which is a bit distracting. Line 199, Line 263 use two different notations $F$ to describe the multi-headed attention mechanism, while the attention is denoted as M in eq. 8b. The notations could be unified for readability.

(5) A lot of the terms in the experiments, for example DS1-8, VIMCO estimator, MrBayes, are wordings that are not familiar to general audience in the ML literatures. The authors should consider expand the appendix section to include some introduction on the technical components, instead of simply citing the terms from the literatures.

**Questions:**

(1) Is there any explanations about why for in Figure 4 ARTreeFormer converges slower than SBN and ARTree at the beginning? What seem to be difficult to learn for ARTreeFormer, and can we attribute this phenomenon to simplification of attention-based architecture?

(2) Similar to the (1) in the weakness section, can the authors provide some justifications of why ARTreeFormer under-perform the ARTree?

(3) The multi-sample lower bound and annealing schedule play significant roles in training. Have the authors tested different schedules or multi-sample configurations? What are the observed effects on the model’s performance?

(4) How sensitive is ARTreeFormer/ARTree ’s performance to different orders of pre-specified leaf nodes?

---

> ### Author Response · Authors · 2024-11-23
> **Official comment by authors (1/2)**
>
> We highly appreciate your constructive review and positive rating of our paper. Here are our responses:
>
> ## Weaknesses
>
> **(1)**: Table 1 reveals that ARTreeFormer consistently lags behind ARTree in terms of KL divergence. (...)
>
> **Response to (1)**: Thanks for this constructive question. First, we want to clarify that ARTreeFormer is mainly designed to improve the computational efficiency of ARTree, rather than the approximation accuracy. To inspect which component contributes to the reduced performance of ARTreeFormer, we conducted an ablation study. We see that the strong representation power of topological node embedding is the most crucial part of a model's performance. Notably, the attention-based local message passing improves both accuracy and efficiency compared to GNN used in ARTree, showing the additional benefit of attention-based modeling for tree topologies.
>
> *Table: KL divergence obtained by various model architectures on the TDE task.*
> | Model | DS1 | DS2 | DS3 | DS4 |
> | --- | --- | --- | --- | --- |
> | Topological Node Embedding + Global GNN ( = ARTree) | 0.0045 | 0.0097 | 0.0548 | 0.0299 |
> | Topological Node Embedding + Local Attention | 0.0089 | 0.0102 | 0.0441 | 0.0276 |
> | Learnable Node Embedding + Global GNN | 0.0044 | 0.0123 | 0.0916 | 0.0346 |
> | Learnable Node Embedding + Local Attention (= ARTreeFormer)| 0.0067 | 0.0102 | 0.0777 | 0.0320 |
>
> **(2)**: It seems that the recurrent node embedding with simplified attention mechanism and (...)
>
> **Response to (2)**: Thanks for the suggestion! For the ablation study on only changing one of the modules of ARTree, please check our response to Weakness (1).
> For the ablation study on different attention mechanisms, we investigate an additional attention mechanism, i.e., linear attention, and report the KL divergences in the following table.
> We see that linear attention can deteriorate the inference accuracy, although it may offer more efficient computation.
>
> *Table: KL divergence obtained by different attention types on the TDE task.*
> | Attention type \ Dataset | DS1 | DS2 | DS3 | DS4 |
> | --- | --- | --- | --- | --- |
> |softmax |0.0067|0.0102|0.0777|0.0320|
> |linear |0.0078|0.0108|0.0884|0.0406|
>
> **(3)**: There is not a theoretical analysis on how ARTreeFormer improves over ARTree in terms of running time.
>
> **Response to (3)**: Thanks for this constructive suggestion.
> The following table compares the computational complexity of the two major modules in ARTree/ARTreeFormer, where $\alpha\in(0,1)$ refers to the accelerated complexity order of linear (1-order) vectorized operations in PyTorch. We have added this discussion in our revision.
>
> We would like to emphasize that $\alpha$ can be small in practice (i.e., fast computation of batched tensors in PyTorch). Even in such a case, the complexity of ARTree's node embedding module is still higher than $O(N^2)$, while those of other modules are reduced to approximately equal to or less than $O(N)$. This validates the observation that the topological node embedding dominates the computation time.
>
> |Module|ARTree|ARTreeFormer|
> |---|---|---|
> |Node embedding module|$O(N^3)$|$O(N^2d+Nd^2)$|
> |Message passing module|$O(N^2d^2)$|$O(Nd^2)$|
> |Node embedding module (vectorization)|$O(N^{2+\alpha})$|$O(N^{1+\alpha}d^{\alpha}+Nd^{2\alpha})$|
> |Message passing module (vectorization)|$O(N^{1+\alpha}d^{2\alpha})$|$O(Nd^{2\alpha})$|
>
> **(4)**: Small thing on the writing, but the authors spend two paragraphs explaining (...)
>
> **Response to (4)**: Thanks for your suggestion! We have now merged the paragraphs for VBPI score and parsimony score together in our revision.
> We would like to clarify that Line 196 and Line 256 are deliberately distinguished by us. 'graph' is for graph-level pooling and 'message' is for local message passing.
> They represent two distinct modules in ARTreeFormer (although they are both implemented by MHA).
>
> **(5)**: A lot of the terms in the experiments, for example DS1-8, VIMCO estimator, MrBayes, are wordings that are not familiar to general audience in the ML literatures.
>
> **Response to (5)**: We apologize for the frequent usage of abbreviations.
> - DS1-8 refers to eight benchmark datasets that are used to evaluate phylogenetic inference problems.
> These data sets consist of collections of benchmark biological sequences, and we aim at inferring trees from them.
> These data sets have been cited in the second paragraph of Section 4.
> - MrBayes is an MCMC software for Bayesian phylogenetic inference and is often considered as a golden criterion in this field. We added a citation in the second paragraph of Section 4.
> - VIMCO is a gradient estimator for multi-sample lower bounds and is not related to phylogenetic inference. We added a description of VIMCO in Appendix C in our revision.

---

> > ### Author Response · Authors · 2024-11-23
> > **Official comment by authors (2/2)**
> >
> > ## Questions
> >
> > **(1)**: Is there any explanations about why for in Figure 4 ARTreeFormer converges slower than SBN and ARTree at the beginning? (...)
> >
> > **Response to (1)**: SBN learns fastest because it uses pre-computed trees for parameterization and initialization, while ARTree and ARTreeFormer do not.
> > In our opinion, ARTreeFormer learns slower than ARTree because the powerful topological node embedding may capture the tree topology information rapidly.
> > This can also be verified by the ablation study of different component combinations in our response to Weakness (1).
> >
> > **(2)**: Similar to the (1) in the weakness section, can the authors provide some justifications of why ARTreeFormer under-perform the ARTree?
> >
> > **Response to (2)**: Thanks for this question. We have provided some empirical justifications in our response to Weakness (1).
> >
> > **(3)** The multi-sample lower bound and annealing schedule play significant roles in training. (...)
> >
> > **Response to (3)**: We agree that the annealing schedule and multi-sample configuration may affect the results.
> > The choices in our paper simply follow previous works [1,2,3,4].
> > Moreover, for the multi-sample configuration, [4] suggested a moderate number of particles because this benefits both mode discovery and posterior approximation, and [2] had tried different number of particles in experiments.
> >
> > **(4)**: How sensitive is ARTreeFormer/ARTree ’s performance to different orders of pre-specified leaf nodes?
> >
> > **Response to (4)**: Thanks for your question! The leaf node order of ARTreeFormer is the same as ARTree [1], and Figure 3 of [1] shows that the performance is not sensitive to the orders of the leaf nodes.
> >
> >
> > [1] Tianyu Xie and Cheng Zhang. ARTree: A deep autoregressive model for phylogenetic inference. In Thirty-seventh Conference on Neural Information Processing Systems, 2023\
> > [2] Cheng Zhang and Frederick A Matsen IV. Variational Bayesian phylogenetic inference. In International Conference on Learning Representations, 2019\
> > [3] Cheng Zhang. Improved variational Bayesian phylogenetic inference with normalizing flows. In Neural Information Processing Systems, 2020\
> > [4] Cheng Zhang and Frederick A Matsen IV. A variational approach to Bayesian phylogenetic inference, JMLR, 2024

---

> > > ### Comment · Reviewer_5tHT · 2024-11-27
> > >
> > > Thank you for the thorough comments.
> > > All my concerns are addressed and I would like to keep the current score for acceptance.

---

### Author Response · Authors · 2024-11-24
**Global response**

We thank all reviewers for the constructive feedback.
We have revised the paper, and have incorporated their suggestions with the following major changes:
- In section 2, we merge the introduction of phylogenetic likelihood and the parsimony score to keep a more compact writing.
- We add a discussion about the computational complexity in Table 1, Section 3.2, and Appendix B.2, which validate that the node embedding module dominates the computation time.
- We added an explanation of the standard deviations of MLL in Section 4.3 and discussed the possibility of using mixture techniques to further improve the results.
- We substitute the qualitative description of acceleration with quantitative ones if applicable. The word "parallelization" is changed to "vectorization" throughout this paper.
- We added the details about the VIMCO estimator in Appendix C.
- We added the time consumption of GeoPhy in Table 8.

---

### Comment · Area_Chair_KDHE · 2024-11-25
**Last day for reviewers to ask questions to the authors!**

Dear reviewers,

Tomorrow (Nov 26) is the last day for asking questions to the authors. With this in mind, please read the rebuttal provided by the authors, as well as the other reviews. If you have not already done so, please explicitly acknowledge that you have read the rebuttal and reviews, provide your updated view _accompanied by a motivation_, and raise any outstanding questions for the authors.

**Timeline**: As a reminder, the review timeline is as follows:
- November 26: Last day for reviewers to ask questions to authors.
- November 27: Last day for authors to respond to reviewers.
- November 28 - December 10: Reviewer and area chair discussion phase.

Thank you again for your hard work,

Your AC

---

### Author Response · Authors · 2024-12-04
**Global Response**

Dear AC and Reviewers,

During the discussion phase, the reviewers raised several concerns regarding the novelty of ARTreeFormer and the claim of improved computational efficiency, and we feel it is necessary to clarify some of them here.

- **Our novelty and contribution**: By leveraging learnable topological node embedding and local message passing, ARTreeFormer overcomes the large time consumption bottleneck of ARTree while maintaining ARTree's favorable approximation accuracy. Its main methodological contribution can be summarized in the following aspects: (i) the learnable node embedding with attention mechanism in ARTreeFormer can be vectorized over different nodes and tree topologies, which reduces the time complexity by one order and has been verified empirically. (ii) Global message passing is substituted by an attention-based local message passing in ARTreeFormer, which can also reduce the time complexity of an order without sacrificing approximation accuracy. (iii) Compared to the GNNs in ARTree, the attention mechanism in the message-passing step of ARTreeFormer is capable of capturing the long-distance dependencies. The following ablation study shows the advantage of the attention mechanism. Attention mechanism for phylogenetic tree modeling itself is an interesting topic and we take the first step in this direction.

*Table: KL divergence obtained by different message passing modules on the TDE task.*
| Model | DS1 | DS2 | DS3 | DS4 |
| --- | --- | --- | --- | --- |
| Learnable Node Embedding + GNN-based global message passing | **0.0044** | 0.0123 | 0.0916 | 0.0346 |
| Learnable Node Embedding + Attention-base local message passing (= ARTreeFormer)| 0.0067 | **0.0102** | **0.0777** | **0.0320** |

- **Complexity analysis**. We provide the time complexity analysis for ARTree and ARTreeFormer, for both cases (with and without vectorized computation). It should be emphasized that with vectorization, the time complexity of ARTree's node embedding module is always 1-order higher than that of ARTreeFormer.
The main reason is that ARTreeFormer enables efficient vectorization across different nodes on a tree, while ARTree does not due to Dirichlet energy minimization for topological node embedding.
Note that although Dirichlet energy minimization can be done in linear time ($O(n)$) for each subtree, this operation needs to be repeated for a sequence of subtrees during the autoregressive generation process of the tree topology.
On the other hand, in ARTreeFormer, for each of the subtrees, we can reduce the complexity to $O(1)$ using learnable node embedding.

- **Computation time baseline**. As suggested by Reviewer 3iPi, the topological node embedding can be calculated via the fix-point algorithm with the square trick.
We agree that this can reduce the computation time of ARTree's node embedding module, but is still much slower (around 6x computation time in our experiments) than that of ARTreeFormer.
We will add this important baseline in our revision.
We also want to point out that although the adjacency matrix is sparse, its $n$-power becomes more dense as $n$ increases, so the acceleration technique for sparse matrix multiplication may not apply for large $n$.

- **About the C++ implementation**. It is undoubted that C++ runs much faster than Python, and we apologize for not being able to deliver a runtime comparison in C++ (this requires rewriting our code base in C++) between ARTree and ARTreeFormer given this short time budget for rebuttal. However, we think that the current comparison in Python is reasonable as explained below.
Based on the profiling of ARTree's node embedding module, a large amount of time (around $55\\%$) is spent on the PyTorch-based numerical computation, which should have a similar consumption in C++.
This is to say, although the overhead of the pure Python part makes the current implementation of ARTree slow, the main part of the node embedding module in ARTree (which is written using Pytorch that has C++ under the hood) can not be sped up by using C++.
Therefore, the current results in our manuscript (Figure 2, Figure 3, Figure 4) are sufficient to show the superior efficiency of ARTreeFormer.

---

### Meta-Review · Area_Chair_KDHE · 2024-12-19

**Metareview:**

This paper proposed a faster model than previous work, called ARTree, for phylogenetic inference. Several reviewers agreed that decreasing runtimes for Bayesian phylogenetics is an important objective by itself.

During the author-reviewer discussions, several reviewers raised the importance of including a complexity analysis of the proposed work, which should explain why the proposed model is faster than the original ARTree. While the authors have responded with a complexity analysis in their rebuttal, these contained notations to include the effect of vectorized operations that were unfamiliar to all reviewers and the AC, and unfortunately further explanations by the authors did not clarify this sufficiently.

Furthermore, reviewer 3iPi had extensive and lively discussions with the authors on the fairness of comparison against previously (supposed inefficient) implementations in python, profiling of the bottlenecks in speed of the proposed work and the original ARTree work, and the subsequent impact of the proposed improvement. Reviewer 3iPi suggested that the improved performance in speed based on a very inefficient implementation in python of the prior ARTree work is not fair. After a significant back and forth between authors and the reviewer on supposed factors of speed up by moving code from python/PyTorch  to pure C++ implementations, the authors conclude that they estimate that a C++ implementation will give only a very limited speed up the main part of the node embedding module in ARTree. However, reviewer 3iPi has provided several arguments against this, and has taken the extensive effort to provide a C++ implementation that runs around 30x faster than the author's Python implementation.
During the AC-reviewer discussions, all reviewers who engaged with the discussion and read the review of 3iPI have agreed that reviewer 3iPi raised valid concerns with ample evidence to back up their claims, and indicated to want to lower their scores.

Given this strong consensus I recommend to reject this paper. I'd like to thank the authors for actively engaging in the discussion phase and for their effort in trying to address all concerns. Furthermore, thank you to all reviewers for engaging in discussion, and in particular thank you to reviewer 3iPi for going far above and beyond what is normally expected and observed from a reviewer.

**Additional Comments On Reviewer Discussion:**

See above.

---

### Decision · Program_Chairs · 2025-01-22

Reject